# Cross-Modal Domain Adaptation for Cost-Efficient Visual Reinforcement Learning

**Xiong-Hui Chen**[*], **Shengyi Jiang**[*], **Feng Xu, Zongzhang Zhang**[†]**, Yang Yu**
National Key Laboratory of Novel Software Technology
Nanjing University, Nanjing 210023, China
Pazhou Lab, Guangzhou 510330, China
{chenxh, jiangsy, xufeng}@lamda.nju.edu.cn, {zzzhang, yuy}@nju.edu.cn

## Abstract

In visual-input sim-to-real scenarios, to overcome the reality gap between images rendered in simulators and those from the real world, domain adaptation, i.e., learning an aligned representation space between simulators and the real world, then training and deploying policies in the aligned representation, is a promising direction. Previous methods focus on same-modal domain adaptation. However, those methods require building and running simulators that render high-quality images, which can be difficult and costly. In this paper, we consider a more cost-efficient setting of visual-input sim-to-real where only low-dimensional states are simulated. We first point out that the objective of learning mapping functions in previous methods that align the representation spaces is ill-posed, prone to yield an incorrect mapping. When the mapping crosses modalities, previous methods are easier to fail. Our algorithm, **C**ross-m**O**dal **D**omain **A**daptation with **S**equential structure (CODAS), mitigates the ill-posedness by utilizing the sequential nature of the data sampling process in RL tasks. Experiments on MuJoCo and Hand Manipulation Suite tasks show that the agents deployed with our method achieve similar performance as it has in the source domain, while those deployed with previous methods designed for same-modal domain adaptation suffer a larger performance gap.

## 1 Introduction

Reinforcement learning (RL) for vision-based robotic control tasks has achieved remarkable success in recent years [1, 2]. However, current RL algorithms necessitate a substantial number of interactions with the environment, which are costly both in time and money on real robots. An appealing alternative is to train policies in simulators, then transfer these policies to real-world systems [3]. Due to inevitable differences of representation between simulators and the real world, which is also known as the "reality gap" [4], applying policies trained in one domain directly to another almost surely fail, especially in visual-input tasks, which is due to the poor generalization of RL policies [5]. Domain adaptation is a promising direction to handle the gap by mapping representation from two domains to an aligned representation and then training and deploying policies in the aligned representation.

Many recent works, which learn a mapping function to align the data distributions of the two domains, have adopted unsupervised visual domain adaptation [6, 7, 8]. We point out that, as illustrated in Fig. 1(a), the objective is ill-posed for learning a correct mapping function. These adaptation methods exploit structural constraints [9] in two domains of the same modality (e.g., learned on simulated **images** and deployed on real **images**). These methods implicitly alleviate the intrinsic ill-posedness of distribution matching.

---

[*]Equal Contribution [†]Corresponding Author

35th Conference on Neural Information Processing Systems (NeurIPS 2021).

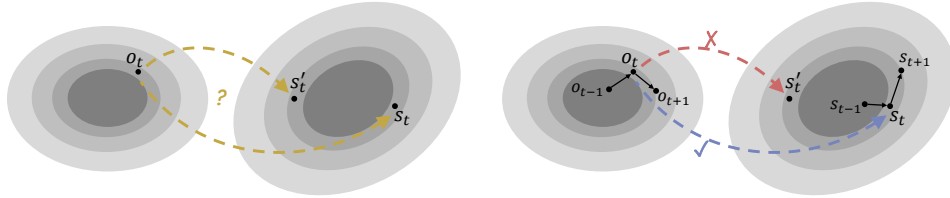

(a) Mapping only considering state-distribution matching

(b) Mapping with sequential structure

Figure 1: Illustration of the training objective of learning a mapping function in unsupervised domain adaptation. Shaded regions denote data distributions, where the darker the color, the higher the probability. For each figure, the left region is the target domain and the right is the source domain. In Fig. 1(a), both $s_t$ and $s'_t$ are "realistic" instances, but only $s_t$ is correct. Since they are of similar probabilities, mapping an instance $o_t$ in the target domain to somewhere of a similar probability in the source domain is "reasonable" if we only consider distribution matching. In RL, the policy may output unreliable actions when taking these incorrectly mapped states as inputs. In Fig. 1(b), a sequential structure can help rule out the wrong mapping via trajectory contexts.

However, such a kind of same-modal domain adaptation requires the simulator to render images when training the model, which introduces unwanted costs and difficulties that are ignored in previous works. First, building a rendering engine is a laborious task. Second, using RL methods to train a policy with an image-based simulator is usually harder [10] and slower (can be up to $20\times$ slower [11]) than with a state-based simulator. An ideal solution to these problems is to train policies with states in the simulator and adapt the learned policies to real-world images. Current domain adaptation methods generally fail in this setting since the structural constraints based on the modality consistency are no longer available, which makes the representation alignment task harder.

To learn such a cross-modal mapping, we propose **C**ross-m**O**dal **D**omain **A**daptation with **S**equential structure (CODAS) that learns a mapping function from images in the target domain to states in the source domain, as illustrated in Fig. 1(b). With the help of the learned mapping function, policies trained on states in the source domain can be deployed in the target domain of images directly. Specifically, based on the sequential nature of RL problems, we formulate the cross-domain adaptation problem as a variational inference problem and decompose it into a series of solvable optimization objectives. We also design a special residual model structure in the recurrent neural network (RNN) to enforce additional inductive bias and stabilize the training process.

We evaluate our method on six MuJoCo [12] tasks and four Robot Hand Manipulation tasks [13], where we treat states as the source domain, and rendered images as the target domain. Results show that the learned mapping function can help transfer the policy to the target domain with only a small performance degradation in most of the tasks while existing methods [14, 15] suffer from a larger performance gap.

## 2 Related Work

Unsupervised visual domain adaptation (UDA) aims to map the source domain and the target domain to an aligned distribution without pairing the data. UDA is originally designed for image translation in computer vision [14, 6, 16, 17]. In RL, UDA transfers policies from simulators to the real world, to overcome the reality gap between rendered images and real-world images. Previous methods fall into two major categories: feature-level adaptation, where domain-invariant representations are learned [7], and image-to-image adaptation, where pixels from source images are used to generate images that look like those from the target domain [18].

Feature-level adaptation usually adopts domain randomization techniques for domain-invariant representation learning [19, 20, 21, 22, 23]. In domain randomization, a meta-simulator is required to generate a mass of variants on rendered images. The learner aims to extract an invariant representation from the variants of representations. Therefore, the training process is generally costly. Besides, these methods implicitly assume that the variants can cover the target domain so that the target domain is only an instance of the variants.

Image-to-image adaptation is challenging when data from two domains are unpaired. Most previous works attempt to solve this problem by using generative adversarial networks (GANs) [15, 24, 3, 17, 25, 26]. [24] transfers policies from Atari games to their modified variants by training a GAN to map images from the target domain to the source domain. RL-CycleGAN [3] unifies the learning of a CycleGAN [14] and an RL policy, claiming better performance by learning features that are most crucial to the Q-function in RL.

Despite the success of the image-to-image domain adaptation methods in previous settings, we point out that these methods can somewhat bypass the ill-posedness for distribution matching, which is illustrated in the example in Fig. 1. Since images generally differ only locally in color, textile, and lighting, but resemble globally between two domains, the effect of ill-posedness can be implicitly handled by leveraging the consistency of modality [17, 25, 26]. However, since images and states differ essentially, we can no longer utilize the consistency of modality. Thus, as shown in our experiments, the effect of ill-posedness is revealed in RL tasks. Some works impose extra structural constraints [9, 27] but fail in image-to-state domain adaptation either. In our work, we force the mapped states to follow transition consistency by considering the inner relationship between the sampled data from the decision-making process in RL tasks.

# 3 Cross-modal Domain Adaptation with Sequential Structure

Without loss of generality, the visual domain adaptation problem in RL can be formulated as learning a mapping function to align the representation of the source and target domains, then training and deploying policies in the aligned representation. To learn the mapping function and policies, a dataset is pre-collected by some policies (e.g., human-expert policies or random policies) in the target domain, and an agent can only interact with the environment in the source domain. Since our major concern is the reality gap between the representation spaces, in this formulation, we follow the setting in previous methods which omits the gap between the dynamics models in the two domains.

In our setting of cross-modal domain adaptation, the source domain is a low-dimensional state domain and the target domain is a high-dimensional image domain. In this work, we regard the state space as the aligned representation space. We first pre-train a policy $\pi$ in the source domain. By learning a mapping $q_\phi$, parameterized by $\phi$, from image space $O$ to state-space $S$, the agent can be deployed with a new policy $\tilde{\pi}(o) = (\pi \circ q_\phi)(o)$, where $\circ$ denotes function composition, i.e., image observations are mapped to states, then the pre-trained policy directly acts on the states.

## 3.1 Domain Adaptation in RL as Variational Inference

In this section, we first define our objective of domain adaptation in RL based on the framework of variational inference. Then we give a comparison between our objective and the objectives used in previous methods.

The modeling of the generation and inference process of the decision process in the source and target domains used in our method is illustrated in Fig. 2. In the source domain, an initial state follows the distribution $p(s_1)$. A transition function $s_t \sim p(s \mid s_{t-1}, a_{t-1})$ outputs the current state $s_t$ from the previous state $s_{t-1}$ and the previous action $a_{t-1}$. In the target domain, the generation process also follows the transition of states, but the states are not available to the agent, and the agent only observes the images corresponding to the states. Ideally, an image observation $o_t$ is generated by the corresponding state $s_t$. In other words, there exists a decoder $p(o_t \mid s_t)$ that can construct observations from states. It is because that the state contains full of information for a transition function to compute the next state, while image observations are rendered based on the states. However, there would have some irrelevant patterns in images in real-world applications. In practice, our decoder is defined as $p(o_t \mid s_t, o_{t-1})$. We allow $p_\theta(o_t \mid \cdot)$ to be dependent on $o_{t-1}$ so that some irrelevant patterns in images can be decoded easily in an auto-regressive manner. The inference process can be regarded as the mapping process in domain adaptation in RL. The inference function is defined as $q_\phi(s_t \mid s_{t-1}, a_{t-1}, o_t)$, which is similar to the transition function but with additional $o_t$ as an input. A simple insight is that the inputs $s_{t-1}$ and $a_{t-1}$ give information to predict the distribution of $s_t$, as the transition function $p(s \mid s_{t-1}, a_{t-1})$ does, and $o_t$ is used to determine the state sample $s_t$ from the distribution.

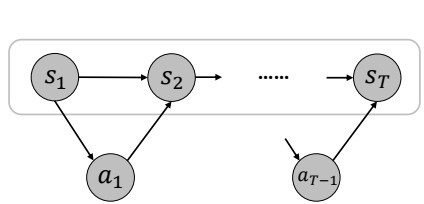 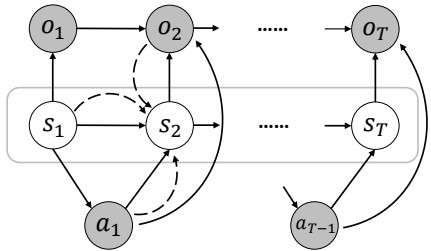

(a) Generation process in the source domain

(b) Generation and inference process in the target domain

Figure 2: Illustration of the generation and inference processes based on the decision processes of RL tasks. All nodes are random variables. Shaded nodes are observable variables. Solid lines denote the generation process and dashed lines denote the inference process. There is a correspondence between the two trajectories enclosed by the rounded rectangles. Note that we include policy $\pi$ in both generation processes, which correspond to the edge from state $s_t$ to action $a_t$.

The training process aims at learning an inference function (i.e., the mapping function) that approximates the ground-truth posterior distribution. The raw optimization objective of variational inference in this setting can be formulated as:

$$\min_{\phi} \mathbb{E}_{\tau^o} \left[ D_{\mathrm{KL}} \left[ q_\phi(\tau^s \mid \tau^o) \mid\mid p(\tau^s \mid \tau^o) \right] \right], \tag{1}$$

where $\tau$ denotes a trajectory, the superscripts $s$ and $o$ of $\tau$ indicate the trajectories from the source and target domains respectively, $p(\cdot)$ is the ground-truth distribution, $q_\phi(\cdot)$ is the mapping function that we want to learn, and $D_{\mathrm{KL}}$ computes the Kullback-Leibler divergence. $\tau^o$ contains a trajectory of observation-action pairs $\{(o_1, a_1), (o_2, a_2), ..., (o_T, a_T)\}$, and $\tau^s$ contains a trajectory of state-action pairs $\{(s_1, a_1), (s_2, a_2), ..., (s_T, a_T)\}$. Here we assume the trajectories of source and target domains are collected by the same policy $\pi$, which is a mild assumption and has been implicitly or explicitly introduced in previous works [24]. Since the policy is fixed, for the simplification of notations, we omit the dependence of $\pi$ and just use $q_\phi$ and $p$ to indicate the distribution. The Evidence Lower BOund (ELBO) of this variational problem can be formulated as:

$$\max_{\phi, \theta} \mathbb{E}_{\tau^o} \left[ \mathbb{E}_{\hat{\tau}^s \sim q_\phi(\tau^s \mid \tau^o)} \left[ \log p_\theta(\tau^o \mid \hat{\tau}^s) \right] - D_{\mathrm{KL}} \left[ q_\phi(\tau^s \mid \tau^o) \mid\mid p(\tau^s) \right] \right], \tag{2}$$

where $\hat{\tau}^s$ denotes the inference trajectories of $q_\phi$, and $p_\theta(\tau^o \mid \hat{\tau}^s)$ is an approximation of the generation process $p(\tau^o \mid \tau^s)$, with parameter $\theta$. The derivation of ELBO can be found in Appendix A. The first term maximizes the reconstruction probability, in order to enforce that the mapped state $\hat{s}$ can recover observation $o$. The second term enforces the alignment of the distributions of the mapped trajectories $q_\phi(\tau^s \mid \tau^o)$ and the ground-truth trajectories distribution $p(\tau^s)$ in the source domain.

Compared with previous works which learn the mapping function by minimizing the divergence between the distributions of $q_\phi(s \mid o)$ and $p(s)$, we model the optimization objective as a matching of trajectory distributions via the Bayesian graphical model. As illustrated in Fig. 1, the trajectory-level modeling is important since the data points are not i.i.d., since they are collected in the RL scenario. Without considering the inner structure of the dataset, mismatching of mapping might occur easily.

## 3.2 Differentiable Optimization Objectives

The ELBO defined in Eq. (2) contains terms involving distributions over the entire trajectory, which is impractical to solve. Given the previously defined generation process, we can decompose the joint probability into the multiplication of single-step probabilities. The result of the decomposition is

$$\max_{\phi, \theta} \mathbb{E}_{\tau^o \sim \mathcal{D}^o} \left[ \sum_{t=1}^{T} \mathbb{E}_{\hat{s}_t \sim q_\phi(s_t \mid \hat{s}_{t-1}, a_{t-1}, o_t)} \left[ \log p_\theta(o_t \mid \hat{s}_t, o_{t-1}, a_{t-1}) \right] - D_{\mathrm{KL}} \left[ q_\phi(\tau^s \mid \tau^o) \mid\mid p(\tau^s) \right] \right], \tag{3}$$

where $\mathcal{D}^o$ denotes the pre-collected dataset in the target domain, $t$ is the timestep and $T$ denotes the horizon. $\hat{s}_0$, $a_0$ and $o_0$ are initialized with $\mathbf{0}$. $q_\phi(s_t \mid \hat{s}_{t-1}, a_{t-1}, o_t)$ is an RNN that outputs the mean

and standard deviation of a Gaussian distribution. A direct computation of the second $D_{\text{KL}}$ term in Eq. (3) is intractable. Following the idea that the optimization process of GAN is equivalent to minimizing a certain distance measurement between two distributions [28], we use the optimization objective of GAN as the alternative of minimizing $D_{\text{KL}}\left[q_\phi\left(\tau^s \mid \tau^o\right) \| p\left(\tau^s\right)\right]$. For the simplicity of implementation and stability of training, we use the original optimization objective of GAN, which is equivalent to minimizing the Jensen-Shannon divergence. The optimization objective is

$$\min_\phi \max_\omega \mathbb{E}_{\tau^s \sim \mathcal{D}^s}\left[\log\left(D_\omega(\tau^s)\right)\right] + \mathbb{E}_{\tau^o \sim \mathcal{D}^o, \hat{\tau}^s \sim q_\phi(\tau^s|\tau^o)}\left[\log\left(1 - D_\omega(\hat{\tau}^s)\right)\right]. \tag{4}$$

Similarly, we decompose Eq. (4) into single-step formulation:

$$\min_\phi \max_\omega L_D(\omega, \phi) = \min_\phi \max_\omega \mathbb{E}_{\tau^s \sim \mathcal{D}^s}\Big[\sum_{t=1}^{T} \log D_\omega(s_t, a_t, h_{t-1})\Big]$$

$$+ \mathbb{E}_{\tau^o \sim \mathcal{D}^o}\Big[\sum_{t=1}^{T} \mathbb{E}_{\hat{s}_t \sim q_\phi(s_t|\hat{s}_{t-1}, a_{t-1}, o_t)} \log\left(1 - D_\omega(\hat{s}_t, a_t, h_{t-1})\right)\Big], \tag{5}$$

where $\mathcal{D}^s$ denotes the dataset collected by the pre-trained policy in the source domain. Here $D_\omega$ is also implemented as an RNN, in which $h$ denotes the hidden state in the RNN rollout. In particular, the discriminator outputs a tuple $(y_t, h_t) = D_\omega(s_t, a_t, h_{t-1})$, where $y_t$ is the probability of prediction and $h_t$ is the next hidden state. We omit the output of $h$ for brevity. The latter term is the practical objective of the KL-divergence in Eq. (3) of the mapping function $q_\phi$.

Combining both the aforementioned reconstruction loss and generation loss together, the optimization objective of the mapping function is

$$\max_{\phi,\theta} \mathbb{E}_{\tau^o \sim \mathcal{D}^o}\Big[\sum_{t=1}^{T} \mathbb{E}_{\hat{s}_t \sim q_\phi(s_t|\hat{s}_{t-1}, a_{t-1}, o_t)}\big[\log p_\theta(o_t \mid \hat{s}_t, o_{t-1}, a_{t-1})$$

$$-\lambda_D \log\left(1 - D_{\omega^*}(\hat{s}_t, a_t, h_{t-1})\right)\big]\Big], \tag{6}$$

$$\text{s.t. } \omega^* = \arg\max_\omega L_D(\omega, \phi),$$

where $\lambda_D$ is the hyper-parameter for the weight controlling. The decoder $p_\theta$ and the discriminator $D_\omega$ are fixed during the training process of the mapping function $q_\phi$. The loss function of the decoder $p_\theta$ is the first term in Eq. (6). Similarly, the mapping function $q_\phi$ is fixed during the training process of the decoder $p_\theta$ and discriminator $D_\omega$. The detailed derivation can be found in Appendix A.

### 3.3 Embedded Dynamics Model for Stable Training

Trajectories of practical RL problems often last hundreds of time steps. Training RNN on such long-horizon trajectories can be difficult. The previous success of ResNet [29] shows that a residual structure can simplify the learning target to predict small residuals, resulting in a remarkable performance increase in learning ultra-deep neural networks.

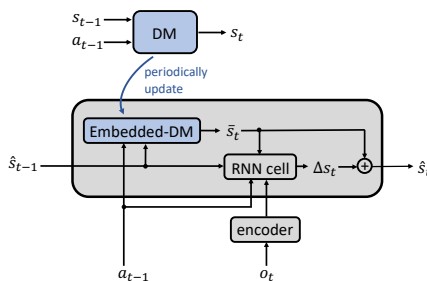

Adopting a similar idea, we incorporate a residual structure in the RNN to help stabilize the training process. A deterministic Dynamics Model (DM) $p_\varphi(s, a)$ with parameter $\varphi$ is trained independently using transition tuples collected in the simulator. Its parameters are periodically updated to embedded-DM. DM minimizes the mean-square error to the transition tuples in the source domain. We regard the

Figure 3: Model structure of the inference function with embedded-DM.

embedded-DM as a part to provide an "average" estimation of the next states $\bar{s}_t$ (see Fig. 3). The job of the rest of the part, instead, is simplified to just output a correction. With this specially designed model structure, the mapped states follow the transition dynamics in the source domain better. In particular, we model the mapping function as:

$$\hat{s}_t = p_\varphi(s_{t-1}, a_{t-1}) + \alpha \Delta s_t, \quad \Delta s_t \sim q_\phi(\Delta s \mid s_{t-1}, a_{t-1}, o_t), \tag{7}$$

where $\alpha$ is a hyper-parameter to control the correction range of $\Delta s$. $\Delta s$ takes tanh as the activation function. Thus, the range of $\alpha \Delta s$ is constrained to $[-\alpha, \alpha]$.

DM is first trained using batches of transition tuples collected in the simulator. However, these data are insufficient for DM training since the DM predictions in unseen state-action pairs are unreliable, due to the extrapolation error [30, 31]. Since the $\alpha \Delta s$ is constrained to $[-\alpha, \alpha]$, too larger extrapolation errors of $p_\varphi(s_{t-1}, a_{t-1})$ might obstruct $\Delta s_t$ to recover the correct $\hat{s}_t$. During the training process of the mapping function, the dynamics model is trained online using $\mathcal{D}^{\hat{s}} = \{(\hat{s}, \pi(a \mid \hat{s}), p(\hat{s}, \pi(a \mid \hat{s})))\}$. That is, we reset the simulator to the mapped states $\hat{s}_t$ and then rollout with a single-step oracle transition to get $s_{t+1}$. The optimization objective of the DM is to minimize the following mean-squared-error loss:

$$\min_\varphi \mathbb{E}_{(s,a,s') \sim \mathcal{D}^s \cup \mathcal{D}^{\hat{s}}}[(p_\varphi(s, a) - s')^2]. \tag{8}$$

When training the mapping function $q_\phi$, the parameter $\varphi$ in the embedded-DM is fixed and gradients are back-propagated from $\hat{s}$ in Eq. (7) to $\phi$ via the reparameterization trick [32]. The detailed training procedure of CODAS is shown in Alg. 1 in the Appendix.

## 4  Experiments

We evaluate our method in MuJoCo [12] from OpenAI Gym and Robot Hand Manipulation Tasks. We define the rendered images as the target domain and the original states as the source domain. The pre-collected dataset in the target domain contains 600 trajectories and is collected by policies trained by PPO [33] and DAPG [34]. Please refer to Appendix E for the detailed experimental setting. The quantitative performance of the data-collecting policies are given in Appendix F.1.

We compare our method with several state-of-the-art methods in same-modal domain adaptation which are also compatible with the cross-modal setting, including:

1. GAN: Conditional GAN trains a mapping function $p_\phi(s \mid o)$ to align the distributions of $\mathcal{D}^o$ and $\mathcal{D}^s$. A discriminator tries to discriminate the mapped states from those from $D^s$, and is used to guide the training of the mapping function.

2. CycleGAN: CycleGAN shares the same basic framework with GAN with an extra cycle-consistent loss to constrain the mapping. In other words, CycleGAN generates $\hat{o}$ from $\hat{o} \sim p_{\phi'}(o \mid p_\phi(s \mid o)))$ and trains the mapping function via making the $D(\hat{o})$ output the real labels with high probability.

3. GAN with Stacked Input: Inspired by a common trick that uses stacked images as input in visual RL tasks [35, 36], we modify GAN to take stacked images as input, and name it GAN STACK. The stacked images provide additional local temporal information.

We also compare CODAS with behavior cloning (BC), which trains a policy in a supervised manner using $(o_t, a_t) \sim \mathcal{D}^o$. All domain-adaptation baselines are trained with the same neural network structure and tuned to converge (i.e., the output probabilities of discriminators are close to 0.5). The tasks are trained for 10,000 to 40,000 epochs based on their difficulty. For each epoch, every method is updated using a batch of 20 trajectories. Implementation details for all these methods can be found in Appendix D and the source code is available at https://github.com/xionghuichen/codas.

To focus on the performance of the adaptation process, we use reward ratio as the metric: $r_{\text{ratio}} = \frac{r}{r^*}$, where $r$ and $r^*$ are the cumulative rewards of the adapted policy deployed on images the optimal policy trained on states respectively. Each task is trained with three seeds. In the following plotted curves, the solid lines indicate the mean value and the shaded regions indicate the range of $\pm 1$ standard deviation. All the details of training and evaluation are given in Appendix E and Appendix F.

### 4.1  Performance on MuJoCo Tasks

The training curves of all methods in MuJoCo tasks are given in Fig. 4. Firstly, BC performs well in HalfCheetah, but poorly in the other environments. This indicates that the dataset is not large enough to learn a correct mapping from images to actions directly. In Swimmer and HalfCheetah, the deployed policies with mapping functions learned by GAN and CycleGAN achieve good performances. It is because of the stability of the environments. The trajectory will not pre-terminate no matter how badly the agent behaves. Therefore, the agent is allowed some incorrect actions, which come from

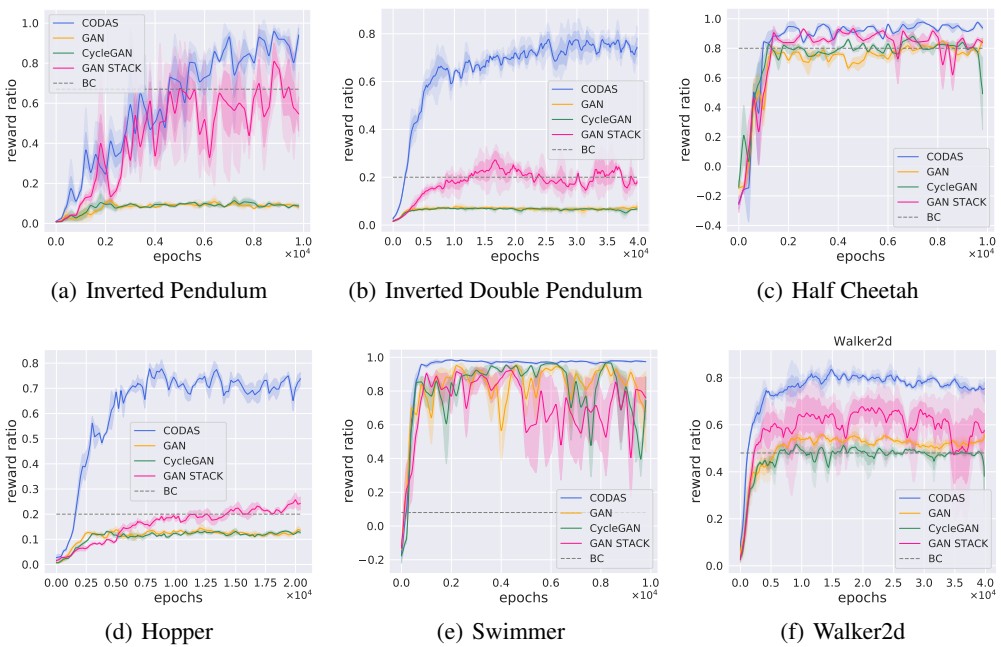

| | | |
|---|---|---|
| (a) Inverted Pendulum | (b) Inverted Double Pendulum | (c) Half Cheetah |
| (d) Hopper | (e) Swimmer | (f) Walker2d |

Figure 4: Training curves of different methods on MuJoCo.

mismatched states. While in the rest tasks, the agent might reach unsafe states after performing some undesired actions. Thus the agent is sensitive to the mismatching of mapping states. In these tasks, GAN and CycleGAN perform even worse than BC. Although CycleGAN has been tuned to converge (see Appendix), the performance of CycleGAN is similar to GAN in all of the tasks. We believe although the cycle-consistent loss guarantees the mapping function is a bijective mapping, the mismatching problem shown in Fig. 1(a) can not be handled, which is more critical in the RL setting. Therefore the performance improvement is not significant. Finally, we found that the GAN STACK algorithm reaches better performances than GAN in most of the tasks. The phenomenon indicates that sequential information is important to learn a correct mapping. However, without other constraints that CODAS has, the performance improvement is unstable.

In conclusion, the performance of CODAS is consistently better than the baseline algorithms. The average performance of adapted policies in the target domain is about 85% of their original performance in the source domain. A visual illustration of the mapped states' accuracy is shown in Fig. 5. Both reconstructed images and re-rendered images match the original ones well. Re-rendered images can even match the original ones well in the last falling frames which are sparse in the dataset.

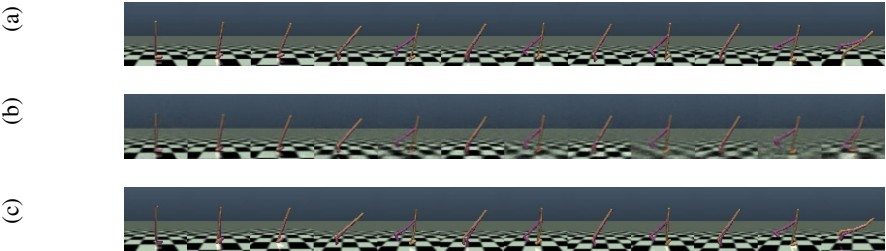

Figure 5: A visual illustration of (a) original images, (b) reconstructed images, and (c) re-rendered images of the mapped states.

## 4.2 Quantitative Results of State Mapping Error

In our experiments, since each image is rendered based on its corresponding state, we can store the state as the ground-truth state for each image in our experiments. For each batch of trajectories $\tau^o$ in an epoch, we can compute the root mean squared error (RMSE) between the ground-truth states and

the mapped states. The curves of the RMSE between the ground-truth states and the mapped states are shown in Fig. 6.

We regard the RMSE as the quantitative results of the mapped states for each algorithm. The results show that CODAS is consistently better than all of the baseline algorithms among all of the tasks. In HalfCheetah and Swimmer tasks, on which the baseline algorithms can also reach a good reward ratio, the qualities of mapped states of CODAS are significantly better. The results verify our suppose in the main body that the good reward ratios come from the robustness of environments to some incorrect output actions, which are caused by mismatched mapped states.

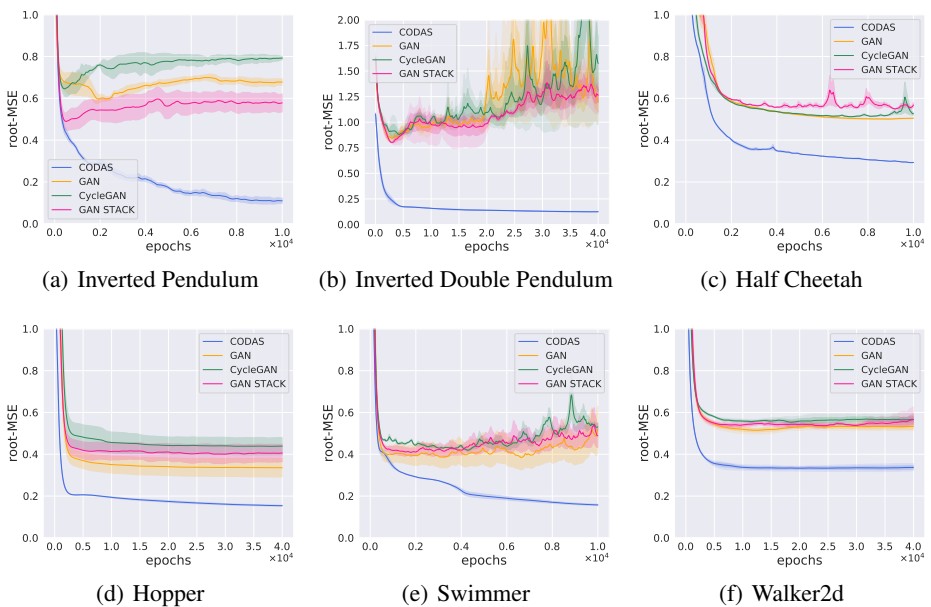

Figure 6: Root mean squared error between mapped states and ground-truth states. The solid lines denote the mean value. The shadows denote the standard deviation.

## 4.3 Ablation Studies

By considering three mapping function modeling methods (i.e., Multilayer Perceptron (MLP), RNN or RNN with embedded-DM (DM-RNN)) and two discriminator modeling methods (i.e., taking trajectories as inputs or not), we construct six CODAS variants to do our Ablation studies. We named them with the format of "mapping method-Y/N". For example, DM-RNN-Y is the algorithm that the mapping function is modeled with RNN and embedded-DM, and the discriminator taking trajectories as inputs, which is the original CODAS. We conduct experiments in Hopper, Inverted Double Pendulum, and Walker2d. We show the results in Fig. 7.

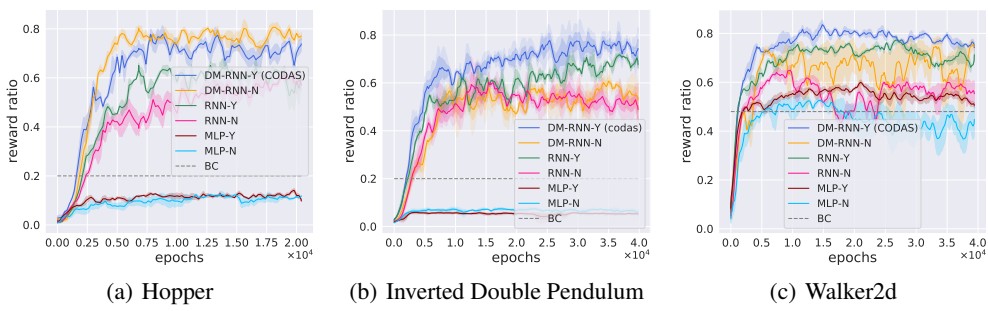

Figure 7: Ablation studies of CODAS.

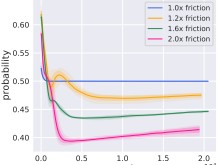 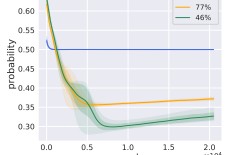 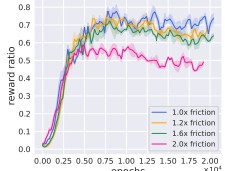 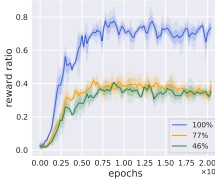

(a) Prediction of discriminator under dynamics mismatch

(b) Prediction of discriminator under policy mismatch

(c) Performance under dynamics mismatch

(d) Performance under policy mismatch

Figure 8: The training curves of CODAS under two types of mismatches. Y-axis in Fig. 8(a) and Fig. 8(b) is the mean of predicted probabilities of the discriminator given a set of trajectories with mapped states. The higher probability means the data are more like to be real. In the dynamics mismatch setting (Fig. 8(a) and Fig. 8(c)), we use "number-x" to denote the rescaling coefficient to the original friction coefficient. For example, "1.2x" denotes the friction coefficient in the source domain is 1.2 times to the target domain. In the policy mismatch setting (Fig. 8(b) and Fig. 8(d)), the percentage is to denote the performance ratio of the policy for data collection compared with the data-collecting policy used in the previous experiments. For example, the legend "77%" means the performance of the data-collecting policy in this experiment has 23% performance degradation.

The comparison of `RNN-*` methods and `MLP-*` methods in Fig. 7 shows that RNN structure is an important component for mapping function training in Hopper and Inverted Double Pendulum tasks. By taking sequential information into consideration, the actions taken in the target domain are more accurate because the policy acts on the mapped states. Based on the RNN mapping function, feeding the sequential information into the discriminator further improves the performance for InvertedDouble and Walker2d. These results verify our argument that the inner structure of the collected dataset is crucial for modeling the universal domain-adaptation objective in the RL setting. On the other hand, in these tasks, embedded-DM helps improve the performance both from `RNN-N` to `DM-RNN-Y` and from `RNN-Y` to `DM-RNN-Y`. Both embedded-DM and the sequential-input discriminator improve the performance of CODAS. Embedded-DM standardizes the inference process which simplifies the learning complexity. The sequential-input discriminator strengthens the gradient signals back-propagated from the discriminator by taking more information into consideration to discriminate the two datasets.

### 4.4 Robustness to Mismatches

Current UDA methods omit the gap between the dynamics models and the data-collecting policies of the source and target domains [17, 25]. Although we follow these implicit consistency assumptions in CODAS, in some real-world applications, the reality-gap of dynamics models is non-negligible [19, 37, 38, 39], and the policies in the two domains might not be the same. Unfortunately, whether considering the reality-gap of dynamics models or the mismatching in the policies, the primitive objective of CODAS and previous UDA methods are biased: Even with an oracle mapping function $p(s \mid o)$, the distribution of the dataset in source domain $\mathcal{D}^s$ and the mapped dataset $\mathcal{D}^{\hat{s}} := \{\hat{s} \mid \hat{s} \sim p(s \mid o), o \in \mathcal{D}^o\}$ are not aligned [40]. In this section, we analyze the effect of the violation on the implicit consistency assumptions in the current UDA paradigm.

We first change the friction coefficient in the Hopper environment to analyze the effect of dynamics mismatches on CODAS. As shown in Fig. 8(a), with a larger friction difference, the asymptotic probability tends to be smaller. The phenomenon indicates that the gap between dynamics models can lead to the gap between the state-action distributions, which can be distinguished by the discriminator. When the friction is doubled, the gap between state distributions starts to affect the deployment performance of policy (Fig. 8(c)). Then we add Gaussian noise with different variance to the original data-collecting policy to collect datasets and train CODAS to analyze the effect of policy mismatch. The policy mismatch is quantified by the performance degradation. We assume that with large performance degradation, the policy behaviors will be more different. The details of the setting can be seen in Appendix E.1.2. As shown in Fig. 8(b), the mismatch of collection policies also induces the gap in the state-action distribution. The gap is larger compared with that of the dynamics mismatch. Such a larger gap does cause a more significant performance drop of deployed policy.

However, in the experiments with 120% and 160% friction, the mapping function is relatively robust to the mismatch. We think that such tolerance comes from the distributional modeling for each step

of inference. The distribution modeling improves the robustness by considering the uncertainty of the prediction. The performance drop in both experiments suggest us pay more attention to the implicit assumptions not fully discussed in current UDA methods in RL tasks. In Appendix F.4, we further test the tolerance to the mismatch of initial state distribution, which has a similar empirical conclusion.

### 4.5 Performance on Robot Hand Manipulation Tasks

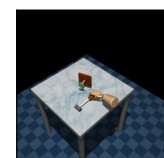

We further test CODAS in four complex hand-manipulation tasks: door, pen, hammer and relocate[2]. An example of image observation of the hammer task is given in Fig. 9. In the hammer task, the robot arm should hold the hammer and hit the nail into the wood. The policies trained by DAPG [34] are used as the pre-trained source domain policy. The performance of CODAS is given in Tab. 1. In three out of four tasks, CODAS yields reasonable mapping functions for policy deployment. CODAS fails to output a correct mapping in the relocate task. We think that this failure can be attributed to the goal-conditional nature of the task, where the robot arm needs to both grasp a ball from and take it to a randomly initialized location in every episode. The goal-conditional nature increases the complexity of data distribution. To learn a better mapping function, a larger dataset is necessary to capture enough information of the distribution.

Figure 9: An example of image observation of the hammer task.

Table 1: The reward ratio in hand-manipulation tasks.

| Tasks | hammer | pen | door | relocate |
|---|---|---|---|---|
| Reward Ratio | 0.820 | 0.701 | 0.886 | 0.090 |

## 5 Discussion and Future Work

In this work, we investigate the cross-modal unsupervised domain adaptation problem in RL that is more cost-efficient than image-to-image domain adaption. We first point out the intrinsic ill-posedness of distribution matching in the current formulation of the UDA objective. To handle this issue, we deduce the UDA objective based on the decision processes of RL tasks and the framework of variational inference, then derive a differentiable objective. We also design a special model structure to stabilize the training and enforce better dynamics consistency. These components constitute our method, Cross-Modal Domain Adaptation with Sequential structure (CODAS).

The experiments in MuJoCo and Robot Hand Manipulation Tasks show the ability of CODAS to find a correct mapping function between two completely different domains with unlabeled data. We hope that CODAS and its reasonable performance could attract and enlighten more work in this setting. Such results also corroborate that modeling the process of generation and inference more precisely is a promising way to improve the performance on UDA, in parallel with previous methods that focus on exploiting more efficient structural constraints. In the ablation studies, we analyze the effectiveness of sequential mapping structure, trajectory-input discriminator, and embedded-DM in CODAS. We also test the robustness of CODAS to two kinds of mismatches, namely dynamics and policy mismatches. The results show that CODAS enjoys a reasonable degree of robustness to both mismatches. However, if the mismatches exceed certain thresholds, the bias of the original objective of CODAS will result in degradation of the deployment performance.

In essence, CODAS tries to formulate and solve the UDA problems in RL without relying on any prior knowledge of the two domains. Image-to-image UDA can be regarded as a special case of cross-modal UDA. Therefore, it is possible to extend CODAS to the image-to-image UDA problems, and other proposed techniques focused on image-to-image UDA to pre-train/constrain a CNN encoder can be adopted into CODAS too, which is valuable to investigate in future work. On the other hand, gaps still exist between our experiments (simulation image to state) and the practical visual sim2real problem (real image to state) in theory. By modeling the mismatching of dynamics models and data-collected policies into the CODAS framework, we can build a more robust UDA algorithm when the assumptions in this paper are not strictly fulfilled. We leave them as our future work.

---

[2]The environments are modified to support resetting to an arbitrary state, which are also open-sourced in https://github.com/jiangsy/mj_envs

## Acknowledgements and Disclosure of Funding

We thank Chenxiao Gao for his feedback on the final presentation of this paper. This work is supported by the National Key Research and Development Program of China (2020AAA0107200), the NSFC (61876077, 61876119), the Fundamental Research Funds for the Central Universities (022114380010), and Huawei Noah's Ark Lab (HF2019105005).

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
