## A Derivation of Optimization Objectives

For a policy $\pi(a|s)$ and a dynamics model $T(s'|s, a)$, its occupancy measure $\rho_\pi^T : \mathcal{S} \times \mathcal{A} \to \mathbb{R}$ can be defined as $\rho_\pi^T(s, a) = \pi(s, a) \sum_{i=0}^{\infty} \gamma^i \Pr\{s_i = s|\pi, T\}$, where $S$ and $A$ denote the state and action space respectively, and $\Pr\{s_i = s|\pi, T\}$ is the probability of $s_i = s$ under policy $\pi$ and dynamics model $T(s'|s, a)$ at time-step $i$ [1]. The occupancy measure can be interpreted as the distribution of state-action pairs that an agent encounters when navigating the dynamics model $T$ with policy $\pi$ [2]. The primitive objective of unsupervised domain adaptation (UDA) is to minimize the data distribution in the source domain and the mapped data distribution from the target domain. From the perspective of policy occupancy measure, it is to learn $p(s|o)$ to minimize

$$D\left[\rho_{\pi^t}^{T^t}(o, a) \,||\, \rho_{\pi^s}^{T^s}(p(s \mid o), a)\right], \forall o \in \mathcal{O}, \forall a \in \mathcal{A},$$

where $\pi^t$ and $\pi^s$ are the data-collecting policies in the target and source domain respectively, $T^t$ and $T^s$ are the dynamics models in the target and source domain respectively, $\mathcal{O}$ is the state space in the target domain (i.e., image observation space in our cross-modal UDA setting), and $D$ is one of a distribution distance measure (e.g., Kullback-Leibler (KL) Divergence).

Since the occupancy measure depends on $T$ and $\pi$, the objective implies two assumptions, which are also the assumptions in our formulations: intuitively, data-collecting policies in the two domains are the same and transition models of the two domains are consistent. Formally, given an oracle mapping function $p^*(s|o)$, $\pi^t(a|o) = \pi^s(a|p^*(s|o)), \forall o \in \mathcal{O}$ and $p^*(s'|T^t(o'|o, a)) = T^s(s'|p^*(s|o), a), \forall o \in \mathcal{O}, \forall a \in \mathcal{A}$ are satisfied. Based on the assumptions, the correct mapping function minimizes $D$ between the state-action distribution in the source domain and the mapped state-action distribution from the target domain.

However, as shown in Fig. 1 (in the main body), a mapping function minimized $D$ between the distributions from the two domains cannot guarantee the mapping function is correct. In this work, we model our objective of domain adaptation based on the decision-making processes of RL to utilize more information inside the state-action distribution. Taking use of the variational inference technique, the optimization objective in this setting can be formulated as:

$$\min_\phi \mathbb{E}_{\tau^o}\left[D_{\mathrm{KL}}\left[q_\phi(\tau^s \mid \tau^o) \,||\, p(\tau^s \mid \tau^o)\right]\right],$$

where $\tau$ denotes a trajectory, the superscripts $s$ and $o$ of $\tau$ indicate the trajectories are from the source and target domains respectively, $p(\cdot)$ is the ground-truth distribution, $q_\phi(\cdot)$ is the mapping function that we want to learn, and $D_{\mathrm{KL}}$ computes the KL divergence. $\tau^o$ contains a trajectory of observation-action pairs $\{(o_1, a_1), (o_2, a_2), ..., (o_T, a_T)\}$, and $\tau^s$ contains a trajectory of state-action pairs $\{(s_1, a_1), (s_2, a_2), ..., (s_T, a_T)\}$.

To maximize the objective, we first transform it into the Evidence Lower BOund (ELBO):

$$D_{\mathrm{KL}}\left[q_\phi(\tau^s \mid \tau^o) \,||\, p(\tau^s \mid \tau^o)\right]$$

$$=\mathbb{E}_{\hat{\tau}^s \sim q_\phi(\tau^s|\tau^o)}\left[\log \frac{q_\phi(\hat{\tau}^s \mid \tau^o)}{p(\hat{\tau}^s \mid \tau^o)}\right]$$

$$=\mathbb{E}_{\hat{\tau}^s \sim q_\phi(\tau^s|\tau^o)}\left[\log q_\phi(\hat{\tau}^s \mid \tau^o) - \log \frac{p(\hat{\tau}^s, \tau^o)}{p(\tau^o)}\right]$$

$$=\mathbb{E}_{\hat{\tau}^s \sim q_\phi(\tau^s|\tau^o)}\left[\log q_\phi(\hat{\tau}^s \mid \tau^o) - \log p(\tau^o, \hat{\tau}^s)\right] + \log p(\tau^o),$$

where $\hat{\tau}^s$ denotes the inference trajectories of $q_\phi$. The last term $p(\tau^o)$ is a constant with regard to the parameter $\phi$, thus can be ignored in the optimization process. The objective can be reduced to minimizing

$$\min_{q_\phi} \mathbb{E}_{\tau^o}\left[\mathbb{E}_{\hat{\tau}^s \sim q_\phi(\tau^s|\tau^o)}\left[\log q_\phi(\hat{\tau}^s \mid \tau^o) - \log p(\tau^o, \hat{\tau}^s)\right]\right].$$

Since the sampled trajectories can be considered as an i.i.d., the first term of expectation can be further re-written as:

$$\mathbb{E}_{\hat{\tau}^s \sim q_\phi(\tau^s|\tau^o)}\left[\log q_\phi(\hat{\tau}^s \mid \tau^o) - \log p(\tau^o, \hat{\tau}^s)\right]$$

$$=\mathbb{E}_{\hat{\tau}^s \sim q_\phi(\tau^s|\tau^o)}\left[\log \frac{q_\phi(\hat{\tau}^s \mid \tau^o)}{p(\tau^o \mid \hat{\tau}^s)p(\hat{\tau}^s)}\right]$$

$$=-\mathbb{E}_{\hat{\tau}^s \sim q_\phi(\tau^s|\tau^o)}\left[\log p(\tau^o \mid \hat{\tau}^s)\right] + D_{\mathrm{KL}}\left[q_\phi(\tau^s \mid \tau^o) \,||\, p(\tau^s)\right].$$

Since $p(\tau^o \mid \hat{\tau}^s)$ is unknown, we use $p_\theta(\tau^o \mid \hat{\tau}^s)$, with parameter $\theta$, as an approximation of the generation process $p(\tau^o \mid \hat{\tau}^s)$. The maximization objective of the variational inference problem becomes:

$$\max_{\phi,\theta} \mathbb{E}_{\tau^o} \Big[ \mathbb{E}_{\hat{\tau}^s \sim q_\phi(\tau^s|\tau^o)} \left[ \log p_\theta \left( \tau^o \mid \hat{\tau}^s \right) \right] - D_{\mathrm{KL}} \left[ q_\phi \left( \tau^s \mid \tau^o \right) \| p \left( \tau^s \right) \right] \Big].$$

The first term maximizes the reconstruction probability, in order to enforce that the mapped state $\hat{s}$ can recover observation $o$. The second term enforces the alignment of the distributions of the mapped trajectories $q_\phi \left( \tau^s \mid \tau^o \right)$ and the ground-truth trajectories distribution $p \left( \tau^s \right)$ in the source domain.

Based on the generation process, we can decompose the first term involving trajectories to multiplication of terms involving states:

$$q_\phi(\tau^s \mid \tau^o) = \prod_{t=1}^{T} q_\phi(s_t \mid s_{t-1}, o_t, a_{t-1}), \ \ o_t, a_{t-1} \sim \tau^o,$$

$$p(\tau^o \mid \tau^s) = \prod_{t=1}^{T} p(o_t \mid o_{t-1}, s_t, a_{t-1}), \ \ s_t, a_{t-1} \sim \tau^s,$$

$$\mathbb{E}_{\tau^o} \Big[ \mathbb{E}_{\hat{\tau}^s \sim q_\phi(\tau^s|\tau^o)} \big[ \log p(\tau^o \mid \hat{\tau}^s)) \big] \Big]$$

$$= \mathbb{E}_{\tau^o} \Big[ \mathbb{E}_{\hat{\tau}^s \sim q_\phi(\tau_s|\tau_o)} \big[ \sum_{t=1}^{T} \log p(o_t \mid \hat{s}_t, a_{t-1}, o_{t-1}) \big] \Big]$$

$$= \mathbb{E}_{\tau^o} \Big[ \sum_{t=1}^{T} (\mathbb{E}_{\hat{s}_t \sim q_\phi(\hat{s}_t|\hat{s}_{t-1}, a_{t-1}, o_t)} \big[ \log p(o_t \mid \hat{s}_t, a_{t-1}, o_{t-1}) \big] \Big].$$

The result of the decomposition is

$$\max_{\phi,\theta} \mathbb{E}_{\tau^o \sim \mathcal{D}^o} \Big[ \sum_{t=1}^{T} \mathbb{E}_{\hat{s}_t \sim q_\phi(s_t|\hat{s}_{t-1}, a_{t-1}, o_t)} \big[ \log p_\theta(o_t \mid \hat{s}_t, a_{t-1}, o_{t-1}) \big] - D_{\mathrm{KL}} \left[ q_\phi(\tau^s \mid \tau^o) \| p(\tau^s) \right] \Big], \tag{1}$$

where $\mathcal{D}^o$ denotes the pre-collected dataset in the target domain, $t$ is the timestep and $T$ denotes the horizon. $\hat{s}_0$, $a_0$ and $o_0$ are initialized with $\mathbf{0}$. $q_\phi(s_t \mid \hat{s}_{t-1}, a_{t-1}, o_t)$ is an RNN which outputs the mean and standard deviation of a Gaussian distribution. A direct computation of the $D_{\mathrm{KL}}$ term in Eq. (1) is intractable. Following the idea that the optimization process of GAN is equivalent to minimizing a certain distance between two distributions [3], we use the optimization objective of GAN as the alternative of minimizing $D_{\mathrm{KL}} \left[ q_\phi \left( \tau^s \mid \tau^o \right) \| p \left( \tau^s \right) \right]$. A new discriminator $D_\omega$, parameterized by $\omega$, is introduced to maximize

$$L_{\mathrm{D}}(\phi, \omega) = \mathbb{E}_{\tau^s \sim \mathcal{D}^s} \Big[ D_\omega(\tau^s) \Big] + \mathbb{E}_{\tau^o \sim \mathcal{D}^o, \hat{\tau}^s \sim q_\phi(\tau^s|\tau^o)} \Big[ - \exp(D_\omega(\hat{\tau}^s) - 1) \Big].$$

The derivation of the objective is based the theorem in f-GAN [3]. Please refer to the Sec. 2 and Sec. 3 in the paper for more details. However, $D_\omega$ is a neural network with a linear activation function, which can become arbitrarily large. For implementation simplicity and training stability, we choose to optimize the original objective of GAN, which is equivalent to minimizing the Jensen-Shannon divergence. The optimization objective is

$$\min_{\phi} \max_{\omega} \mathbb{E}_{\tau^s \sim \mathcal{D}^s} \big[ \log \left( D_\omega(\tau^s) \right) \big] + \mathbb{E}_{\tau^o \sim \mathcal{D}^o, \hat{\tau}^s \sim q_\phi(\tau^s|\tau^o)} \big[ \log \left( 1 - D_\omega(\hat{\tau}^s) \right) \big], \tag{2}$$

where $D_\omega$ is a neural network with a sigmoid activation function. Similarly, we decompose Eq. (2) into single-step formulation:

$$\min_{\phi} \max_{\omega} L_D(\omega, \phi) = \min_{\phi} \max_{\omega} \mathbb{E}_{\tau^s \sim \mathcal{D}^s} \big[ \sum_{t=1}^{T} \log D_\omega(s_t, a_t, h_{t-1}) \big]$$

$$+ \mathbb{E}_{\tau^o \sim \mathcal{D}^o} \big[ \sum_{t=1}^{T} \mathbb{E}_{\hat{s}_t \sim q_\phi(s_t|\hat{s}_{t-1}, a_{t-1}, o_t)} \log \left( 1 - D_\omega(\hat{s}_t, a_t, h_{t-1}) \right) \big], \tag{3}$$

where $\mathcal{D}^s$ denotes the dataset collected by the pre-trained policy in the source domain. Here $D_\omega$ is also implemented as an RNN, in which $h$ denotes the hidden state in the RNN rollout. In particular, the discriminator outputs a tuple $(y_t, h_t) = D_\omega(s_t, a_t, h_{t-1})$, where $y_t$ is the probability of prediction and $h_t$ is the next hidden state. We omit the output of $h$ for brevity. The latter term is the practical objective of the KL-divergence in Eq. (1) of the mapping function $q_\phi$.

Combining both the aforementioned reconstruction loss and generation loss together, the optimization objective of the mapping function is:

$$
\max_{\phi,\theta} \mathbb{E}_{\tau^o \sim \mathcal{D}^o} \Big[ \sum_{t=1}^{T} \mathbb{E}_{\hat{s}_t \sim q_\phi(s_t|\hat{s}_{t-1}, a_{t-1}, o_t)} \big[ \log p_\theta(o_t \mid \hat{s}_t, a_{t-1}, o_{t-1})
$$
$$
- \lambda_D \log \big( 1 - D_{\omega^*}(\hat{s}_t, a_t, h_{t-1}) \big) \big] \Big], \tag{4}
$$
$$
\text{s.t. } \omega^* = \arg\max_\omega L_D(\omega, \phi),
$$

where $\lambda_D$ is the hyperparameter for the weight controlling. The decoder $p_\theta$ and the discriminator $D_\omega$ are fixed during the training process of the mapping function $q_\phi$. The loss function of the decoder $p_\theta$ is the first term in Eq. (4). Similarly, the mapping function $q_\phi$ is fixed during the training process of the decoder $p_\theta$ and discriminator $D_\omega$. Instead of training $\omega$ to converge before training $\phi$ and $\theta$, we use a single-step gradient method [3] to train CODAS, as current GAN-based techniques do. The details of training can be found in Alg. 1.

## B   A Simple Example of Incorrect Distribution Matching

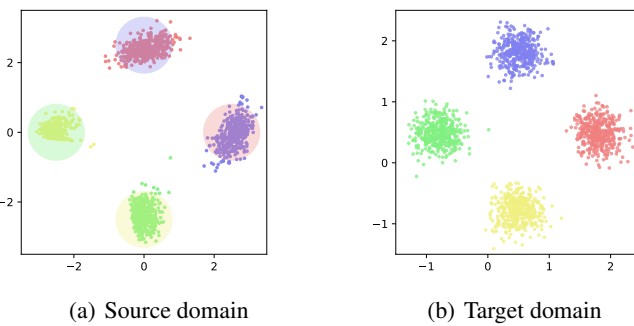

(a) Source domain                    (b) Target domain

Figure B.1: An example of failed distribution matching in GAN. Figures 1(a) and 1(b) are two Gaussian Mixture Models (GMM). The four transparent circles in the left figure denote the source domain, while the small dots in the right figure are samples of the target domain, both indicating a four-component GMM. The small dots in the left figure are the mapped points of a GAN from the target domain, where the color indicates their corresponding points in the target domain. The target domain is achieved by a linear transformation from the source domain, as indicated by the x- and y-axis.

Figure B.1 shows an example of wrong mapping using GAN. A Wasserstein GAN is trained to learn a mapping from the target domain to the source domain. A correct mapping from the target domain to the source domain should map the sampled dots in the target domain to somewhere in the transparent circle of the same color. Due to the nature of GANs, the mapping function only minimizes a distribution distance measure (e.g., Jensen-Shannon divergence) on the distributions of the two domains.

## C   Additional Related work

In RL, representation learning aims to transform high-dimensional observations into lower-dimensional representations. It is widely accepted that learning policies from low-dimensional states are more sample-efficient than learning from pixels, both empirically [4] and theoretically [5].

Recurrent neural network (RNN) is a common neural network structure to learn state representations. Early work on deep RL from images [6] uses a two-step learning process to learn a lower-dimensional state representation. In particular, an auto-encoder is first trained to learn a low-dimensional representation. Subsequently, a policy or model is learned based on this representation. Later work on model-based RL improves representation learning by jointly training the encoder and the dynamics model end-to-end [7], which is effective in learning useful task-oriented representations.

A recent model-based learning method PlaNet [8] learns state embeddings using variational inference, which maps sequential image inputs to a subspace in a low-dimensional vector space to learn a dynamics model more efficiently.

In this work, we also consider the RNN structure for state representation learning and use variational inference for objective modeling. The key difference between our work and previous works comes from the setting of the problem, thus the derived objectives are also different. In previous works, representation learning is used to learn an arbitrary low-dimensional vector representation which is more sample-efficient for downstream training. In our setting, the target representation is pre-defined. A mapping function is trained to minimize the distance between the distributions of the mapped states and the corresponding representation in the simulator.

## D  Detailed Implementations of CODAS

### D.1  Detailed training procedure

The overall computation flow of CODAS at timestep $t$ is illustrated in Fig. D.1. At each timestep $t$, the visual encoder embeds $o_t$ into $e_t$ firstly. The mapping function takes $a_{t-1}$, $e_t$, $\hat{s}_{t-1}$ and $h^s_{t-1}$ as input, where $h^s_{t-1}$ and $\hat{s}_{t-1}$ are the hidden state and the mapped state of the recurrent network at the previous timestep respectively, and outputs $\hat{s}_t$ and the current hidden state $h^s_t$. In the visual decoder, instead of taking $\hat{s}_t$ and $o_{t-1}$ as input, it receives $\hat{s}_t$ and $e_{t-1}$ to avoid duplicate computation. Then, it outputs the reconstructed image $\hat{o}_t$. Similarly, at each timestep $t$, the discriminator receives mapped states $\hat{s}_t$ (or source domain states $s_t$ that are not shown in Fig. D.1), action $a_t$, and hidden states $h^d_{t-1}$ and outputs a label $\hat{y}_t$ to decide where the states come from.

The pseudo-code of the training procedure is provided in Alg. 1.

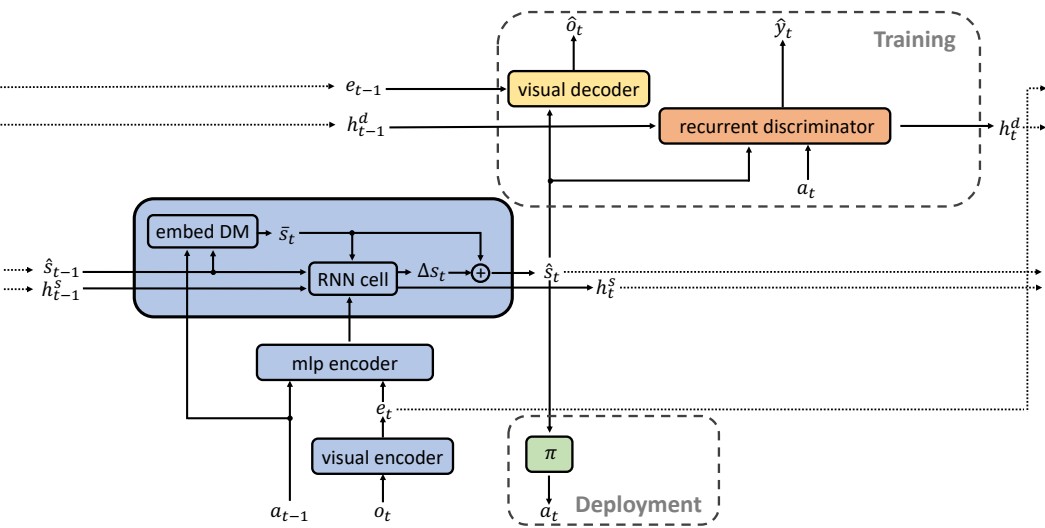

Figure D.1: Illustration of full network structure at timestep $t$. Blue parts denote mapping function $q_\phi$; the yellow part denotes reconstruction function $p_\theta$; the orange part denotes the discriminator $D_\omega$; the green part denotes the pre-trained policy that is fixed during the entire training process.

---
**Algorithm 1** Training Procedure of CODAS
---

**Require:** A simulator with oracle dynamics $p(s' \mid s, a)$; a policy $\pi(a \mid s)$ pre-trained in the source domain; pre-collected datasets $\mathcal{D}^o = \{\tau_1^o, \tau_2^o, ...\}$ and $\mathcal{D}^s = \{\tau_1^s, \tau_2^s, ...\}$ in the target domain and the source domain respectively; the number of iterations $N$; the number of the discriminator updates $I_\mathrm{D}$; the number of the deterministic dynamics model updates $I_\mathrm{DYN}$; the number of the mapping function updates $I_\mathrm{M}$; the number of the reconstruction decoder updates $I_\mathrm{R}$; the frequency of the deterministic dynamics model copied to embedded-DM $F_\mathrm{DYN}$.

1: Initialize a mapping function $q_\phi(\hat{s}_t \mid o_t, a_{t-1}, \hat{s}_{t-1})$, a deterministic dynamics model $p_\varphi(s, a)$ a discriminator $D_\omega(s_t, a_t, h_{t-1})$ and a decoder $p_\theta(\hat{o}_t \mid \hat{s}_t, a_{t-1}, o_{t-1})$.
2: **for** $n = 1$ to $N$ **do**
3:     Sample a batch of trajectories $\{\tau_1^o, \tau_2^o, ...\tau_k^o\}$ from $\mathcal{D}^o$ in the target domain.
4:     For each $\tau_i^o$, infer the state trajectories $((\hat{s}_1, a_1), ..., (\hat{s}_T, a_T))$ using $q_\phi$ to construct the trajectory $\hat{\tau}_i^s$.
5:     **for** $i = 1$ to $I_\mathrm{D}$ **do**
6:         Update the discriminator $D_\omega$ by Eq. (3).
7:     **end for**
8:     **for** $i = 1$ to $I_\mathrm{R}$ **do**
9:         Update the decoder $p_\theta$ by Eq. (4) with fixed $\phi$.
10:     **end for**
11:     Compute the next states for each state-action pair in $\tau_i^s$ to construct the dataset $\mathcal{D}^{\hat{s}} = \{(\hat{s}, \pi(a|\hat{s}), p(\hat{s}, \pi(a|\hat{s}))\}$
12:     **for** $i = 1$ to $I_\mathrm{DYN}$ **do**
13:         Update the deterministic dynamics model $p_\varphi$ by Eq. (8).
14:     **end for**
15:     **if** $n \mod F_\mathrm{DYN} == 0$ **then**
16:         Copy the weight of $p_\varphi$ into the embedded-DM structure in $p_\phi$.
17:     **end if**
18:     **for** $i = 1$ to $I_\mathrm{M}$ **do**
19:         Update the mapping function $p_\phi$ by Eq. (4) with fixed $\theta$ and the embedded-DM structure.
20:     **end for**
21: **end for**

---

## D.2   Implementation Details

In this section, we give several important implementation details in CODAS. All details can be found in our open-source code that will be released after the final decision of this paper.

**Deterministic dynamics model training details**   For each task, 5000 additional trajectories are collected in the source domain (i.e., simulator) to train the deterministic dynamics model. In the tasks of the MuJoCo environment, the collected trajectories are sampled uniformly in the training process. In the tasks of the Robot Hand Manipulation environment, trajectories are sampled by the pre-trained policies[1]. Additional noise is gradually added to the output actions. In particular, the noise is sampled from $U(-\epsilon, \epsilon)$, where $\epsilon$ is increased from $0$ to $0.5$ in the process of data collection. We named the dataset $\mathcal{D}^{s'}$.

We first pre-train the deterministic dynamics model through the dataset $\mathcal{D}^{s'}$. In the main loop of CODAS, when training the deterministic dynamics model, instead of sampling tuples from $\mathcal{D}^s \cup \mathcal{D}^{\hat{s}}$, we sample tuples from $\mathcal{D}^{s'} \cup \mathcal{D}^{\hat{s}}$.

**Trajectory truncation for discriminator**   The trajectory information empowers the discriminator with greater capability to discriminate the inputs from the two domains. Without enough data collected, which might happen in the target domain, the dataset can not capture enough information about the data distribution. The discriminator tends to overfit some unimportant details of the trajectories, which makes the training of the mapping function unstable. In our implementation, the hidden state of the trajectory-input discriminator is periodically reset to alleviate the problem, that is, $h^d$ is periodically reset to zeros. Here we introduce a hyperparameter $H_\mathrm{reset}$ to denote the period.

---

[1]https://github.com/aravindr93/hand_dapg

Empirically, $H_{\text{reset}}$ is set to 10% to 20% of the horizon for each task. The detailed setting can be found in Tab. 1.

**Soft value clipping on $\bar{s}$ and $\hat{s}$**  Large errors on state prediction including $\bar{s}$ and $\hat{s}$ also affect the stability of the training of the mapping function. In our implementation of CODAS and the baseline algorithms, for each output of $\bar{s}$ and $\hat{s}$, we clip it into a range of $[s_{\min} - \Delta, s_{\max} + \Delta]$, where $s_{\min}$ and $s_{\max}$ are the minimal and maximal values in $\mathcal{D}^{s'}$ and $\Delta = \beta(s_{\max} - s_{\min})$ with $\beta = 0.05$ in all of the tasks. The clip operation would also stop the gradient when training. Therefore, we use a soft clip operation in our implementation. In particular, given a state $s$, the soft clip is:

$$s \leftarrow s_{\min} - \Delta + \text{softplus}((s_{\max} + \Delta - \text{softplus}(s_{\max} + \Delta - s)) - s_{\min} - \Delta),$$

where $\text{softplus}(x) = \log(1 + e^x)$. For each output of $\bar{s}$ and $\hat{s}$ in the training and deploying process, the soft clip operation will be followed.

**State Normalization**  The value range of states in the tasks varies. For example, some dimensions have a range larger than 100, while others have a range smaller than 1. The varied range of state space makes the training of the mapping function unstable. For CODAS and the compared baseline algorithms, we normalize the states with the mean and the standard deviation computed from $\mathcal{D}^{s'}$.

### D.3  Hyperparameters

The detailed network structure and hyperparameters are listed in Tab. 1.

Table 1: Hyperparameters for CODAS

| Type | Name | Value (MuJoCo) | Value (Robot) |
|------|------|----------------|---------------|
| **General** | $I_{\text{D}}$ | 1 | |
| | $I_{\text{DYN}}$ | 5 | |
| | $F_{\text{DYN}}$ | 1 | |
| | $I_{\text{R}}$ | 1 | |
| | $I_{\text{M}}$ | 5 | |
| **Dynamics Model** | Hidden layers | [512, 512, 512] | |
| | Activation function | tanh | |
| | Learning rate | $1 \times 10^{-4}$ | |
| | Minibatch size | 1024 | |
| **Discriminator** | Hidden layers | [256, 256, 256, 256, 256] | [2048, 256, 256, 256, 256] |
| | Activation function | tanh | |
| | Learning rate | $5 \times 10^{-5}$ | |
| | RNN type | GRU | |
| | RNN layers | [128] | |
| | $H_{\text{reset}}$ | 100 | 20 |
| **Mapping Function** | **Encoder** | | |
| | Conv layers | Conv (4, 4, 32), Conv (4, 4, 64), Conv (4, 4, 128), Conv (4, 4, 256) | |
| | MLP hidden layers | [256, 256, 256, 256, 256] | |
| | Activation function | leaky relu | |
| | **RNN Cell** | | |
| | RNN type | GRU | |
| | RNN layers | [128, 128] | |
| | $\alpha$ | 1.0 | |

Table 1: Hyperparameters for CODAS (continued)

| Type | Name | Value (MuJoCo) | Value (Robot) |
|---|---|---|---|
| **Mapping Function** | **Visual Decoder** | | |
| | Hidden layers | [256, 256, 1024] | |
| | Deconv layers | Deconv (4, 4, 128), Deconv (4, 4, 64), Deconv (4, 4, 32) | |
| | **Train** | | |
| | $\lambda_D$ | 20 | |
| | Learning rate | $1 \times 10^{-4}$ | $5 \times 10^{-5}$ |
| | Trajectories per iteration | 10 | 20 |

# E   Experimental Setting

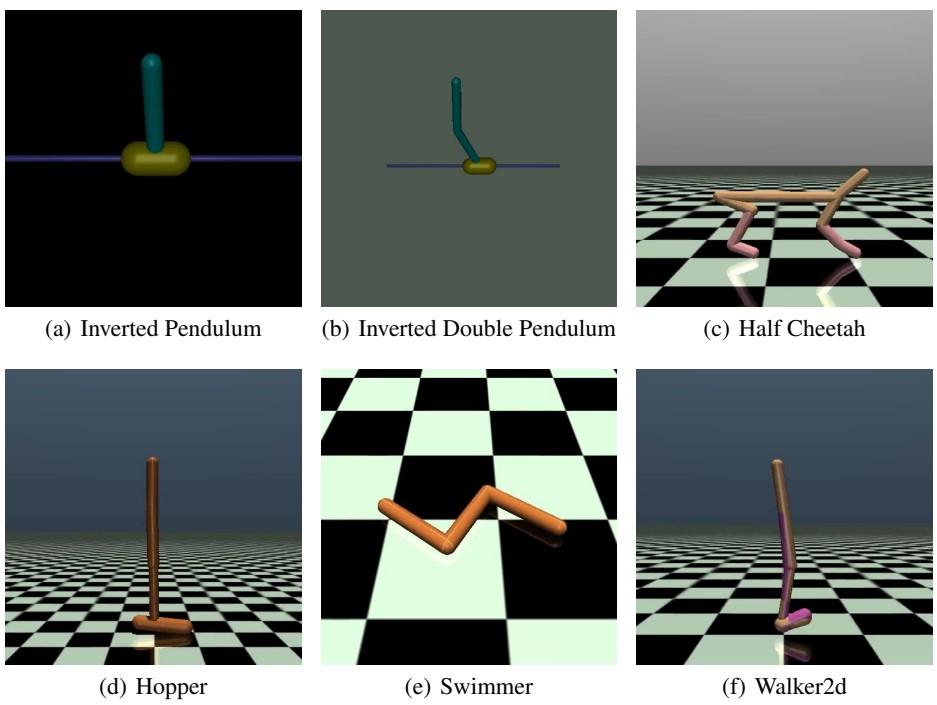

(a) Inverted Pendulum     (b) Inverted Double Pendulum     (c) Half Cheetah

(d) Hopper     (e) Swimmer     (f) Walker2d

Figure E.1: Examples of rendered images of MuJoCo environments

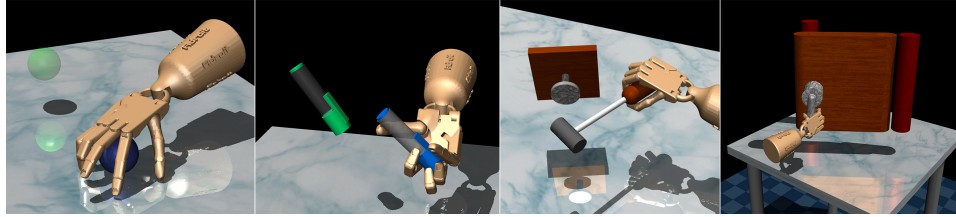

Figure E.2: Examples of rendered images of Robot Hand Manipulation environments

CODAS and baseline algorithms are evaluated in six tasks of the MuJoCo environment [9], including InvertedPendulum, InvertedDoublePendulum, HalfCheetah, Hopper, FixedSwimmer, and Walker2d. We also evaluate our method in four tasks of the Robot Hand Manipulation environment [10], including hammer, door, relocate, and pen. The illustration of these tasks can be found in Figs. E.1 and E.2.

We treat the simulated observations as low dimensional simulation states $s$ and rendered images as real images. The dimension of state and action for each task is listed in Tab. 2.

In HalfCheetah, Hopper, FixedSwimmer, and Walker2d, images are collected by "track camera". In InvertedPendulum and InvertedDoublePendulum, images are collected by "default camera". All images are resized to $[64, 64, 3]$ without any further pre-processing techniques in all tasks. Examples of rendered images are shown in Fig. E.1. In Robot Hand Manipulation tasks, the original states used cannot fully reset the simulator. Thus we modify the state space to a "full-state space", which can fully simulate the next state from any given state. This full-state space is regarded as the state space of the source domain. The mapping function is trained to map the images from the target domain to the full-state space. The images are also rendered via the full states. However, the pre-trained DAPG policies take original states as inputs. For policy deployment, we first map an image from the target domain to a full state, then recover an original state from the mapped full state through the simulator. the recovered state is fed into the pre-trained policy to output an action. The source code of the modified environments will be released along with the source code of CODAS.

Policies are trained via PPO [11] and DAPG [12] for the MuJoCo environment and the Robot Hand Manipulation environment respectively. The policies are regarded as the pre-trained policies in the source domain. For each task, we use the same policy to collect trajectories in the simulator and render the corresponding images for each state. The collected image dataset is regarded as the pre-collected dataset in the target domain. The image dataset contains 600 episodes. In the MuJoCo environment, each episode is truncated to a maximum length of 500 for better computation efficiency. In the Robot Hand Manipulation environment, we keep the original maximum length (i.e., 100 in the pen task and 200 in other tasks). The evaluations of all methods are done based on this truncated dataset.

Table 2: Dimension of state and action for each task

| Task Name | State Dimension | Action Dimension |
|---|---|---|
| Hopper | 11 | 3 |
| Walker2d | 17 | 6 |
| HalfCheetah | 17 | 6 |
| Swimmer | 9 | 2 |
| InvertedPendulum | 4 | 1 |
| InvertedDouble | 11 | 1 |
| pen | 45 | 24 |
| hammer | 46 | 26 |
| relocate | 39 | 30 |
| door | 39 | 28 |

## E.1 Implementation Details of Dynamics Mismatch and Policy Mismatch

### E.1.1 Implementation Details of Dynamics Mismatch

To create dynamics mismatches between the target domain and the source domain, we modify the friction setting in the environment configuration of the Hopper task. Specifically, the friction in the source domain is set to 1.2/1/6/2.0 times the friction in the target domain when creating a dynamics model. When training the mapping function, we regard the simulator with modified friction as the source domain dynamics model and keep the simulator with default friction as the target-domain dynamics model.

### E.1.2 Implementation Details of Policy Mismatch

In the main body, the policy mismatch is quantified by the performance degradation. In fact, the policy mismatch is created via adding noise to the action taken by the data-collecting policy in the target domain. When collecting a dataset in the target domain, for each timestep, we feed the data-collecting policy with the observed state, then the noise $\epsilon_a$ is sampled from a Gaussian distribution $\epsilon_a \sim \mathcal{N}(0, \sigma)$ and added to the action taken by the policy. The noisy action is finally taken by the agent to interact with the environment. Such noisy actions lead to performance degradation of the

policy. The 77% and 46% relative performances correspond to Gaussian distributions of $\sigma = 0.4$ and 0.8, respectively.

# F  Extra Experimental Results

## F.1  Performance of the sub-optimal policies trained with RL in the two domains

Table 3 shows the mean value and standard deviation of the un-discounted cumulative return of 100 trajectories collected by the optimal policy trained on states using PPO. The maximum episode length is set to 1000. Full training curves of policies trained on state space are provided in Fig. F.1. The performance of policies trained on state space matches the public benchmark results. [2] For the MuJoCo environment, policies are trained for 1000000 timesteps using the PPO2 implementation from stable_baselines[3]. For each iteration, 2048 transitions are sampled from the environment, and the policy is updated for 32 epochs with a mini batchsize of 32.

Figure F.1 also provides training curves of the policy trained on images. We modify the network structure of the actor and critic to adapt PPO to image input. In all tasks, the policies perform poorly. The final performance of image input is calculated at $3.0 \times 10^6$ timestep, when the value loss has converged. As far as we know, there is no public performance benchmark of the optimal policy trained on MuJoCo images.

In this work, we argue that using RL methods to train a policy with an image-based simulator is usually harder [4] than with a state-based simulator. The results show that the performances of policies with image input are consistently worse than those with state input. The results demonstrate that the setting of cross-modal domain adaptation is more cost-efficient from the policy training perspective.

Table 3: Performance of the optimal policy on state space

| Tasks | Return |
|---|---|
| Hopper | $2097 \pm 411$ |
| Swimmer | $325 \pm 5$ |
| Walker2d | $3669 \pm 587$ |
| InvertedPendulum | $775.5 \pm 319$ |
| HalfCheetah | $1580 \pm 35$ |
| InvertedDouble | $5201 \pm 1029$ |

As shown in [12], traditional RL algorithms struggle to solve the Robot Hand Manipulation environments directly. We use the policies trained with DAPG [12] as the pre-trained policies. The results of DAPG policies in the four Robot Hand Manipulation tasks can be found in Tab. 4.

Table 4: Performance of DAPG in Robot Hand Manipulation environment

| Task | Return |
|---|---|
| hammer-v0 | $8381 \pm 7471$ |
| pen-v0 | $3275 \pm 1763$ |
| door-v0 | $2916 \pm 431$ |
| relocate-v0 | $4356 \pm 759$ |

## F.2  Training Curves of the Discriminator

For all of the baseline algorithms, we use the same hyperparameters (can be found in Tab. 1) for the discriminator training. At the end of each epoch, we evaluated the averaged predicted probabilities to the mapped states. For each input, the higher probability means the discriminator regards the input as a real sample with a higher probability. We give the predicted results of the discriminator in Fig. F.2.

---

[2]https://spinningup.openai.com/en/latest/spinningup/bench.html
[3]https://stable-baselines.readthedocs.io/en/master/

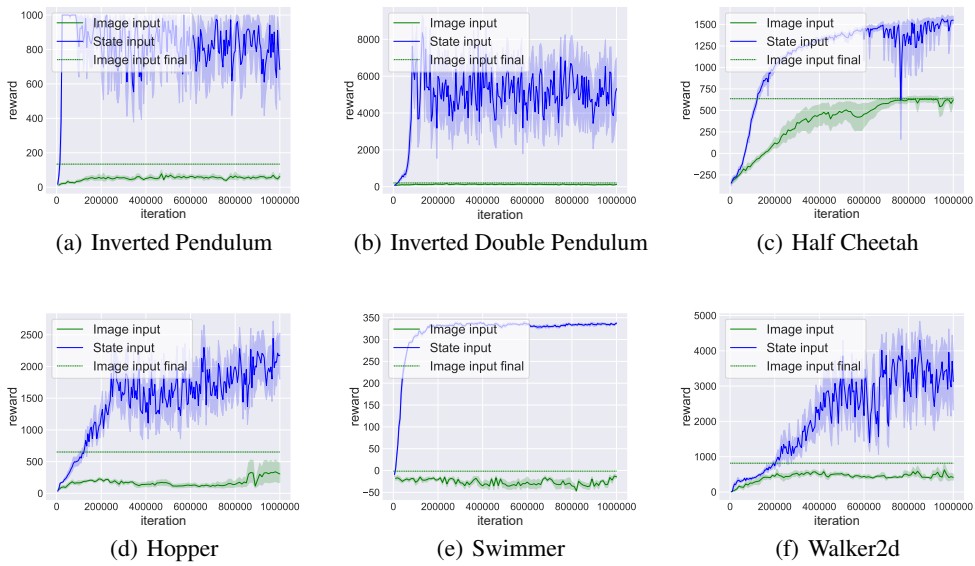

Figure F.1: Training curves of policy on state-space and image-space environments.

As can be found in Fig. F.2, all of the discriminators are converged to near 0.5. Since the discriminators are trained with the same hyperparameters, the results indicate that all the mapping functions have mapped the image data distribution to a state distribution which is indistinguishable for the discriminator to the real state distribution sampled from the simulator. However, the correct distribution matching is not equivalent to the correct image-to-state mapping. Therefore, these methods only considering the distribution mapping is easy to fail to find a correct mapping, while CODAS finds a mapping function with both smaller and more stable RMSE error and has better deployed-policy performance.

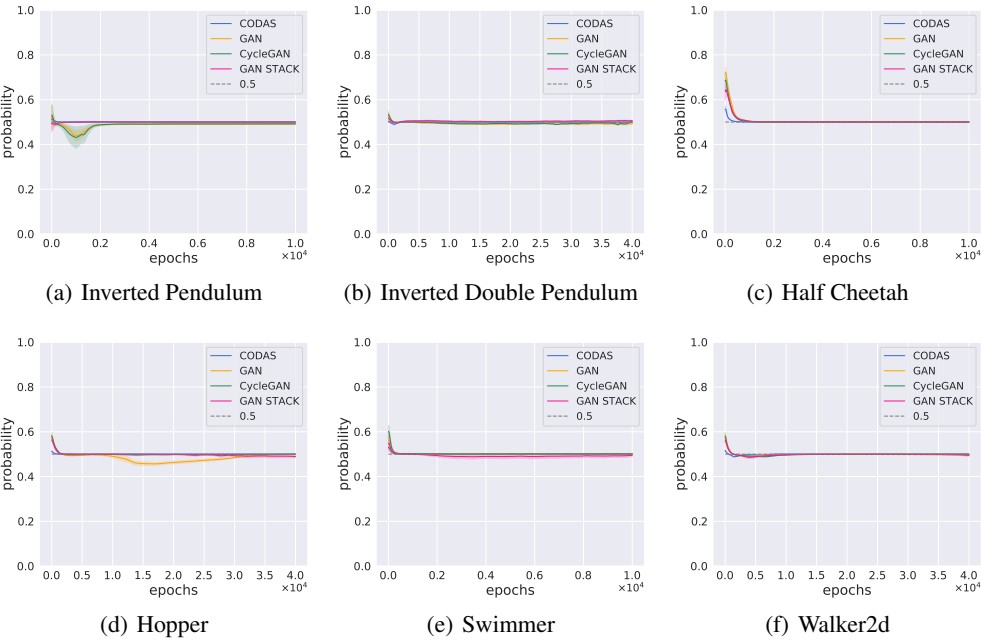

Figure F.2: Predicted probabilities of discriminator on mapped states. The solid lines denote the mean value. The shadows denote the standard deviation.

## F.3 Training Curves of Behavior Cloning

Behavior cloning is simply training the policy network via supervised training. The policy is trained for 200000 iterations, in which a batch of 128 image-action pairs are uniformly sampled from the dataset for each iteration. The dataset is the same as the one used to train CODAS. An adam optimizer with a learning rate being $1e-5$ is used to update the policy network with hidden sizes being [256, 256, 128]. The loss is defined as the mean squared error between the actions taken by the policy network and the policy. The training curves of behavior cloning (BC) are shown in Fig. F.3. The results show BC performs well in HalfCheetah, but is unstable in the other tasks. This indicates that the dataset is not large enough to learn a correct mapping from images to actions directly.

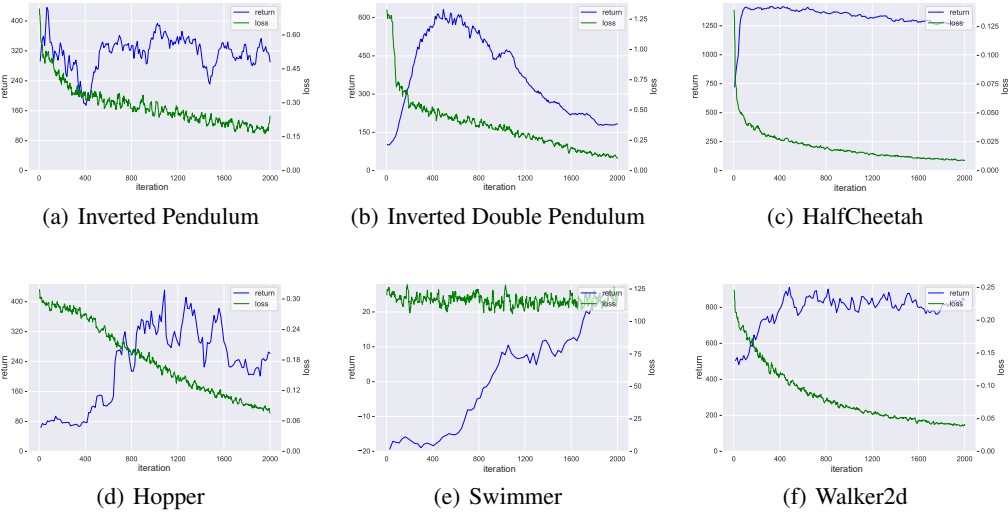

Figure F.3: Training curves of behavior cloning.

## F.4 Robustness to the Initial State Distribution Mismatching

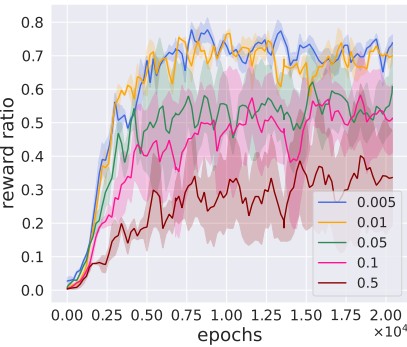

Figure F.4: Illustration of the performance in Hopper task when the initial state distributions are mismatched.

In Hopper tasks, the initial state is generated by a constant state with additional perturbations from a uniform distribution $U(-\alpha, \alpha)$, where $\alpha = 0.005$ in the original tasks.

We further test the robustness (without oracle) to the initial state distribution mismatching by setting the initial inferred state to zero and increasing in the source domain.

As can be seen in Fig. F.4, the performance of CODAS remains unchanged when $\alpha$ is doubled to 0.01. When $\alpha$ is increased to 0.1, CODAS can still reach a reward ratio of around 0.6, which is 82% of the original performance. The results prove that, without an oracle initial state mapping, CODAS is still robust to a stochastic initial state distribution under the current model design. However, if the

initial state is indeed highly stochastic (e.g., $\alpha = 0.5$), an oracle initial state mapping may help the performance.

## F.5 Robustness to the Datasize

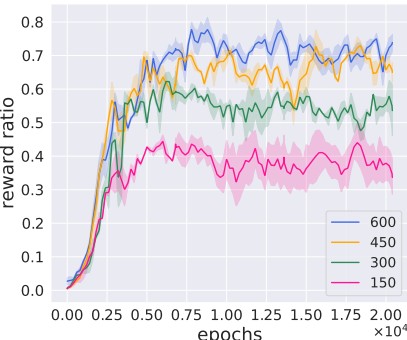

Figure F.5: Illustration of the performance in Hopper task when the size of the dataset is decreased.

We conducted experiments in the Hopper task to test the requirement of CODAS of target domain data for the better integrity of this work. The results can be seen in Fig F.5. In summary, CODAS generally keeps a similar performance when the data size decreases to 450 episodes. When the data size is 300, the reward ratio is still around 0.57. When the data size is 150, the reward ratio drops to 0.4. The results prove that this cross-modal domain adaptation setting and our method CODAS are possible to use fewer real samples when employed as a sim2real approach. However, with too few real samples, the correctness of the mapping function degrades. The results are as expected: To optimize an objective of distribution divergence minimization, it is inevitable to collect enough data to represent a distribution. Without enough data, the gradient given by the discriminator will mislead the training of the mapping function.

## F.6 Extended Results in Robot Hand Manipulation Tasks

The training curves of CODAS in four hand-manipulation tasks are given in F.6. We also show the RMSE and predicted probabilities of discriminator in Fig. F.7 and Fig. F.8 respectively. As shown in Fig. F.6, CODAS yields reasonable mapping functions for policy deployment in three out of the four tasks. The results demonstrate the capacity of the mapping function learned by CODAS to complex environments. However, in Fig. F.8, we found that the predictions of discriminators in these tasks tend to be smaller than 0.5, which contradicts the results in MuJoCo tasks. It means that the discriminator indeed can distinguish distributions of mapped states and simulation states to some degree, but the mapping function can not learn from it. Although the performance of the deployed policies has not been significantly affected in door, pen, and hammer tasks, this phenomenon indicates the mapping function trained with CODAS can not align well from the images space to the state space. It is reasonable since the complexity of learning the mapping function is higher in Robot hand-manipulation tasks because of their larger state space (see Tab.2). CODAS fails to produce a correct mapping in the relocate task, and the prediction probability of the discriminator is the smallest among the four tasks. We think that the failure comes from the goal-conditioned nature of the task, in which the robot arm grasps a ball and takes it to a randomly initialized target location in every trajectory. The goal-conditioned nature further increases the complexity of data distribution. To learn a better mapping function, a larger dataset is necessary to capture enough information of the distribution.

## F.7 Computational Resources

Each CODAS experiment is trained on a single Nvidia GTX 2080Ti. Training CODAS in a MuJoCo tasks takes about 3 days to converge while training CODAS in a Robot Hand Manipulation task takes about 7 days to converge.

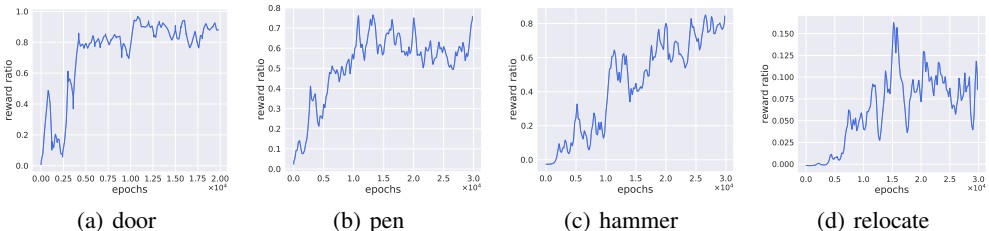

| (a) door | (b) pen | (c) hammer | (d) relocate |

Figure F.6: Reward ratios in Robot Hand Manipulations tasks. The mean value is plotted by the solid lines.

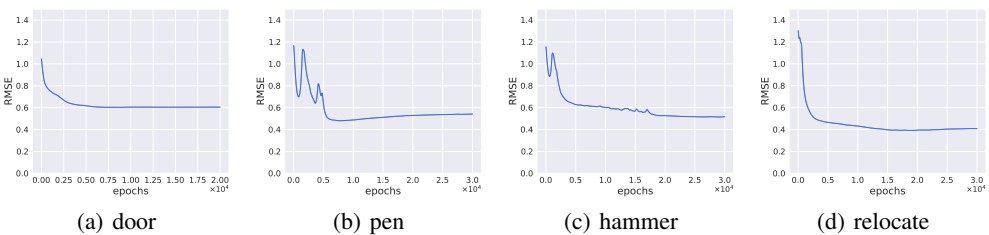

| (a) door | (b) pen | (c) hammer | (d) relocate |

Figure F.7: Root mean squared error between mapped states and ground-truth states in Robot Hand Manipulations tasks. The solid lines denote the mean value.

# G  Further Discussion about cross-modal unsupervised domain adaptation

In this paper, we propose the cross-modal unsupervised domain adaptation setting in RL as a cost-efficient setting for real-world applications. However, here exist practical applications that have a great number of unstructured shapes of 3D objects. In these applications, it is harder to construct a state-based simulator than an image-based simulator. Therefore, in these applications, directly learning from an image-based simulator can be a better choice. On the other hand, there are also many real-world applications that the robots make decisions in enclosed environments. For example, to solve the Rubik's Cube task with a physical robot hand, the state is represented by a vector of the face angles, cube pose, and fingertip locations [13]; to solve the object grasping task with a Baxter robot, the state is represented by a vector of positions of the target object and joint angles of the robotin [14]. In these applications where state space can be easily designed by human experts, the image-to-state UDA pipeline is valuable.

Technique selection should be analyzed case by case. For example, in environments with a simple image-to-action mapping, behavioral cloning of the policy is more efficient. In environments where robots need to face many unstructured objects or unknown obstacles, images can better capture the necessary information to train a policy. It is better to use an image-to-image UDA pipeline to solve the sim2real problems. Based on that, this paper gives a new possibility to solve the sim2real problems, which is suited for the applications that informative state space is easy to design while constructing a renderable simulator and train image-based policies are more costly.

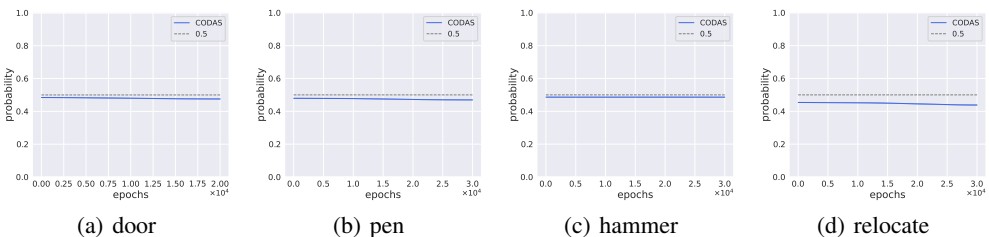

| (a) door | (b) pen | (c) hammer | (d) relocate |

Figure F.8: Predicted probabilities of discriminator on mapped states in Robot Hand Manipulations tasks. The solid lines denote the mean value.