# OpenReview forum: "Cross-modal Domain Adaptation for Cost-Efficient Visual Reinforcement Learning"
_NeurIPS.cc/2021/Conference — NeurIPS 2021 Poster_

### Official Review · Reviewer_6QJy · 2021-07-16

**Rating:** 5
**Confidence:** 4

**Summary:**

This paper purpose to solve sim to real problem from a new viewpoint. Instead of using image or feature level domain adaptation, the authors use sequential information（trajectories) to adapt the image to state, and define a differential and stabilize object for domain adaptation in reinforcement learning, and results perform better than other baselines in all experiments.

**Limitations And Societal Impact:**

The authors have addressed the limitations and potential negative societal impact of their work.

**Main Review:**

Strength:
+ This paper has a solid theoretical derivation for the objective of domain adaptation in RL using Variational Inference.
+ Lots of experiment is conducted to show the effect of this method including baseline, an experiment in a complex task, ablation study, mapping state error, etc.

Weakness:
- Without an image, the agent cannot know the geometry of itself or object which can be hard to train a task need to know the shape (for example grasp for different objects).
- For a simulation environment that has a simple and unrealistic image, it still needs 600 episodes (minimum 100 steps each episodes) data to train the mapping function, which means need more data for the true sim to real.
- This method is proposed to solve sim2real, no experiment show this method can map a real image to the corresponding state, even for a demonstration.

Detail comments：
The main motivation of this paper is for general domain adaptation in sim2real under the same modal. It has a mismatching problem and is costly, so this paper purpose to solve the problem by cross-modal domain adaptation.
For the method, there is some previous work also try to adapt between image and state. For example, in [1], the authors adapt image and state to hidden state and use that to train the policy.  [2] also uses sequential information to unsupervised adapt the policy. It would be better to show the advantage compare with these methods.
For the “cost” problem, the main idea of sim2real is to use a simulator to reduce the cost of real data collection, for this method although it reduces the cost in simulation, it seems to require more complex real data collection than general domain adaptation(trajectory need action). Besides, this paper does not show any demonstration in real-word scenario.  Therefore, it is hard to evaluate whether this method still remains low-cost in real-world scenarios. From this point, it would be better to conduct some experiments in real world or evaluate the performance of this method with different amounts of target data episodes to show some potential for low real data requirement.
For baselines, there is also another domain adaptation method that can be used for cross-modal domain adaptation, such as[1] and [3], it would be better also use this kind of method as stronger baseline.

[1] Wilson, Matthew, and Tucker Hermans. "Learning to manipulate object collections using grounded state representations." Conference on Robot Learning. PMLR, 2020.

[2] Hansen, Nicklas, et al. "Self-supervised policy adaptation during deployment." arXiv preprint arXiv:2007.04309 (2020).

[3] Tzeng, Eric, et al. "Adversarial discriminative domain adaptation." Proceedings of the IEEE conference on computer vision and pattern recognition. 2017.


**Time Spent Reviewing:**

8

---

> ### Author Response · Authors · 2021-08-09
> **Official Response to Reviewer 6QJy**
>
> Thanks for your constructive comments on our paper. Our answers to your questions are as follows:
>
> **1. Without images, is it hard to train a task that needs to know the geometry of itself or object? (e.g., grasp for different objects)**
>
> If information of objects like geometry is necessary for the policy, it can be added to the state space since we can design our simulator to provide sufficient state information.
>
> **2. What is the difference between CODAS and previous work that adapts between image and state [1]?**
>
> [1] maps images of two domains to a hidden state space and an image-based simulator is needed to generate state-image pairs for CNN encoder (i.e., the mapping function) learning. Therefore, the mapping function learning is **supervised** and relies on the high-quality rendered images in the simulator to guarantee the generalization ability of the CNN encoder. We have mentioned such kind of work in the related work section (L64 - 69) and will add this work to the paper.  In CODAS, the state-space is pre-defined by the state-based simulator. When training the mapping function, we do not need a renderable simulator to generate state-image pairs and try to learn the mapping function unsupervisedly.
>
> **3. What is the difference between CODAS and [2] that also uses sequential information to unsupervised adapt the policy?**
>
> [2] maps images of two domains to a hidden state space via $\pi_e$. This state-space is learned rather than pre-defined. $\pi_e$ is learned by policy gradients plus an auxiliary task of inverse dynamics predictions via an inverse dynamics model $\pi_s$. Firstly, it also needs a renderable image-based simulator to learn $\pi_e$ and $\pi_s$, and the generalization ability highly relies on the quality of rendered images. Besides, it needs online samples to achieve adaptation through $\pi_s$. In CODAS, only an offline dataset is needed. In many applications, online sampling with unsafe policies in the target domain is often risky and costly, while the offline dataset collected with safe expert policies is cheap [6, 7, 8].
>
> CODAS is not the first paper to use RNN in RL. RNN is a commonly used tool in RL to encode sequential information as mentioned in Sec. C in Appendix. We have given a detailed comparison there. The contribution of our paper is the motivation why sequential information is needed in this topic, instead of using RNN itself. As shown in previous unsupervised visual domain adaptation (UDA) works in RL [6, 7, 8], the objective of minimizing the divergence of state(-action) distribution is adopted directly with many additional losses to stabilize the mapping function learning. These works are successful in many complex applications. However, the soundness of their objectives is ignored.
>
>
> **4. Does CODAS consider the cost of collecting more complex real data collection than general domain adaptation (trajectory need action)?**
>
> Thanks for bringing up this problem. We will add it to the final section of the paper as a potential improvement direction. Previous work like GAILfO [4] has shown the potentiality of imitating a policy by minimizing the divergence of state distribution of two policies, instead of minimizing that of the state-action distribution like GAIL [5]. We think a similar idea can be adopted to CODAS and will investigate how to relax the requirement of action data in future work.
>
>
> **5. The main idea of sim2real is to use a simulator to reduce the cost of real data collection. For a simulation environment that has a simple and unrealistic image, it still needs 600 episodes of data to train the mapping function, which means need more data for the true sim to real. It is hard to evaluate whether this method still remains low-cost in real-world scenarios. From this point, it would be better to conduct some experiments in real-world or evaluate the performance of this method with different amounts of target data episodes to show some potential for low real data requirement.**
>
>
> In many applications, the motivation of sim2real is to reduce the cost of **online** real data collection, since collecting data with insecure policies is risky and costly. In these applications, **offline** dataset collection with secure policies (such as human expert policies or rule-based sub-optimal policies) is allowed and cheap, and larger-scale data is easier to obtain. We expect that CODAS can be applied in such scenarios [6, 7, 8]. In particular, [6] requires 3,000 episodes of data to achieve 72\% reward ratio and 80,000 episodes of data to achieve 95\% reward ratio. [7] requires 5,000 episodes of on-line collected data to achieve 91\% reward ratio. Some earlier work [8] even requires 500,000 episodes of data. From this point, the size of data collected in the target domain is not a major problem, as long as they are pre-collected by a safe policy.
>
> The value of CODAS is beyond a real-world practice to the RL community and our claim and conclusion in the paper is self-consistent with the current experiment results.
>
> - We propose a new setting of cross-modal domain adaptation in RL. The motivation of cross-modal domain adaptation is to reduce the costs in current visual sim2real problems that require an image-based simulator. First, a rendering engine needs costly human engineering. Second, using RL methods to train a policy with an image-based simulator is usually harder and slower.
>
> - We point out that the distribution minimization objective is ill-posed for UDA in RL. Instead of adopting the distribution minimization objective and GAN directly as previous studies do, our algorithm is focused on rigorous derivations on unsupervised domain adaptation problems based on the framework of variational inference;
>
> - Besides MuJoCo benchmark tasks, we also select hand manipulation tasks, which have high-dimensional state and action space (See Tab. 2  in Sec. E of Appendix), complex context of image space (See Fig. 4 in Sec. E of Appendix), and sophisticated policy behavior, to demonstrate the ability of the mapping function trained by CODAS. On benchmark tasks, we have conducted a full ablation study to the components of CODAS and tolerance study to the assumptions of CODAS to verify the claims in our work.
>
> [1] Matthew Wilson and Tucker Hermans. "Learning to manipulate object collections using grounded state representations." CoRL 2020.
>
> [2] Nicklas Hansen, Rishabh Jangir, Yu Sun, Guillem Alenyà, Pieter Abbeel, Alexei A. Efros, Lerrel Pinto, and Xiaolong Wang.  "Self-supervised policy adaptation during deployment." arXiv:2007.04309.
>
> [3] Eric Tzeng, Judy Hoffman, Kate Saenko, and Trevor Darrell. "Adversarial discriminative domain adaptation." CVPR 2017.
>
> [4] Faraz Torabi, Garrett Warnell, and Peter Stone. "Adversarial imitation learning from state-only demonstrations." AAMAS 2019.
>
> [5] Jonathan Ho and Stefano Ermon. "Generative adversarial imitation learning." NIPS 2016.
>
> [6] Kanishka Rao, Chris Harris, Alex Irpan, Sergey Levine, Julian Ibarz, and Mohi Khansari. "RL-cyclegan: Reinforcement learning aware simulation-to-real." CVPR 2020.
>
> [7] Stephen James, Paul Wohlhart, Mrinal Kalakrishnan, Dmitry Kalashnikov, Alex Irpan, Julian Ibarz, Sergey Levine, Raia Hadsell, and Konstantinos Bousmalis. "Sim-to-real via sim-to-sim: data-efficient robotic grasping via randomized-to-canonical adaptation networks." CVPR 2019.
>
> [8] Konstantinos Bousmalis, Alex Irpan, Paul Wohlhart, Yunfei Bai, Matthew Kelcey, Mrinal Kalakrishnan, Laura Downs, Julian Ibarz, Peter Pastor, Kurt Konolige, Sergey Levine, and Vincent Vanhoucke. "Using simulation and domain adaptation to improve efficiency of deep robotic grasping." ICRA 2018.

---

> ### Author Response · Authors · 2021-08-15
> **Experimental Results of CODAS Under Different Data Sizes in the Target Domain**
>
> We have conducted experiments in the Hopper task to test the requirement of CODAS of target domain data for the better integrity of this work.
> CODAS generally keeps a similar performance when the data size decreases to 450 episodes. When the data size is 300, the reward ratio is still around 0.57. When the data size is 150, the reward ratio drops to 0.4.
> The results prove that this cross-modal domain adaptation setting and our method CODAS are possible to use fewer real samples when employed as a sim2real approach. However, with too few real samples, the correctness of the mapping function is hurt. The results are as expected: To optimize an objective of distribution divergence minimization, it is inevitable to collect enough data to represent a distribution. Without enough data, the gradient given by the discriminator might mislead the mapping function training.
>
> In some applications, the real data are indeed expensive, as the reviewer said. However, we restate that the cross-modal UDA setting and the CODAS method are valuable in many scenarios where offline sampled data is cheap, as long as they are collected by a safe policy, as mentioned in the previous comment. How to further reduce its sample complexity is also interesting. We will investigate it in future work.
>
> The experiment is conducted on Hopper-v2. The original figure can be found in https://postimg.cc/FfR6j1Jc

---

> > ### Comment · Reviewer_6QJy · 2021-09-01
> > **Thanks for the detailed response**
> >
> > Thanks for your detailed response. The response addressed my main concern on the sample efficiency and clarified the relation to previous work. But I do not think that the state of the objects can be added to the state space in an easy way. That is because there are a great number of unstructured shapes of 3D objects in the world, it is infeasible to represent them with a few parameters.  This will limit the useability of the CODAS in complex environments, e.g. manipulating many objects of different shapes or navigating in a room with many obstacles. However, the authors did not discuss this issue. So I just updated my score from 4 to 5.

---

> > > ### Author Response · Authors · 2021-09-01
> > > **Response to Reviewer 6QJy**
> > >
> > > Thanks for the constructive response.
> > >
> > >
> > > First, there indeed exists practical applications that have a great number of unstructured shapes of 3D objects. In these applications, it is harder to construct a state-based simulator than an image-based simulator. Therefore, in these applications, we do agree that directly learning from an image-based simulator can be a better choice.
> > >
> > > However, there are also many real-world applications that the robots make decisions in enclosed environments. For example, in [1], to solve the Rubik’s Cube task with a physical robot hand,  the state is represented by a vector of the face angles, cube pose, and fingertip locations; in [2], to solve the object grasping task with a Baxter robot, the state is represented by a vector of positions of the target object and joint angles of the robot. In these applications where state space can be easily designed by human experts, the image-to-state UDA pipeline is valuable.
> > >
> > > In fact, we do not expect to propose a pipeline to be a universal solution to all sim2real applications. Technique selection should be analyzed case by case.  For example, in environments with a  simple image-to-action mapping, behavioral cloning of the policy is more efficient.
> > > In environments where robots need to face many unstructured objects or unknown obstacles, images can better capture the necessary information to train a policy. It is better to use an image-to-image UDA pipeline to solve the sim2real problems.
> > > Based on that, we give a new possibility to solve the sim2real problems, which is suited for the applications that informative state space is easy to design while constructing a renderable simulator and train image-based policies are more costly.  We think the reviewer’s proposed limitation on image-to-state UDA is valuable and we will add this to the discussion section of the revised paper.
> > >
> > > Second, we think the useability of the image-to-state UDA pipeline does not limit the feasibility of CODAS itself. CODAS is formulated in a general cross-modal UDA setting. Essentially, we try to formulate and solve the UDA problems without relying on any prior knowledge between the two domains. Image-to-image UDA can be regarded as a special case of cross-modal UDA. Therefore, we think it is possible to extend CODAS to the image-to-image UDA problems, and other proposed techniques focused on image-to-image UDA to pretrain/constrain a CNN encoder can be adopted into CODAS too. We think it is interesting and will investigate this in future work.
> > >
> > > We hope that the comment can resolve your concern. Looking forward to your response if you have any further concerns.
> > >
> > >
> > >
> > > [1] Ilge Akkaya, Marcin Andrychowicz, Maciek Chociej and et.al., Solving Rubik's Cube with a Robot Hand, arXiv preprint arXiv:1910.07113 (2019)
> > >
> > > [2] Fangyi Zhang, Jürgen Leitner, Zongyuan Ge and et.al., Adversarial Discriminative Sim-to-real Transfer of Visuo-motor Policies, arXiv preprint arXiv:1709.05746

---

### Official Review · Reviewer_kd5t · 2021-07-17

**Rating:** 6
**Confidence:** 3

**Summary:**

The paper provides a variational formulation to train a observation-to-state mapping, as a potential solution to cross domain transfer. The proposed approach is evaluated in MuJoCo control and hand manipulation tasks, showing its advantage over GAN-based approaches.

**Limitations And Societal Impact:**

Yes.

**Main Review:**

Strength
- The method shows empirical improvement over baselines across a range of control tasks.
- The study of changes in environmental dynamics is very interesting and shows that the proposed method can tolerate some degree of dynamic mismatch.
- While the baselines are rather naive, the paper addresses an interesting problem and proposes a feasible solution.

Weakness
- The method relies on the ability to reset the simulator and query the oracle one-step transition, while the GAN baselines do not have this information. It strongly favors the proposed method. Stacking the images could already improve vanilla GAN significantly, and what if transitional information is also taken into accounts? Requiring the ability to reset could also be a problem when scaling to real robots.

**Time Spent Reviewing:**

2

---

> ### Author Response · Authors · 2021-08-09
> **Official Response to Reviewer kd5t**
>
> Thank you for your appreciation of our paper and your constructive comments to it. Our answers to your questions are as follows:
>
> **1. What is the performance by taking full transition information into account in GAN?**
>
> In our ablation study, "RNN-N" can be regarded as a vanilla GAN with full transitional information. It can improve the performance but is not as stable as CODAS.
>
> **2. Is requiring the ability to reset a problem when scaling to real robots?**
>
> CODAS only requires the ability to reset the source domain (i.e. simulator). We believe this will not be a huge problem when extending CODAS to real robots since we can design simulators that can be reset to a specific state.

---

### Official Review · Reviewer_cBxr · 2021-07-17

**Rating:** 5
**Confidence:** 3

**Summary:**

This paper introduces an interesting work that applies UDA technique to connect the visual gap between the real and simulation domain in RL simulation tasks. An inference function is learned with variational inference to transfer the policy learned from the simulator instance to a realistic instance.


**Limitations And Societal Impact:**

This paper has limited impact on the related community since similar works have been explored.

**Main Review:**

+ This basic idea of this paper is easy to follow.
+ This paper shows an easy solution to reduce the visual gap between real and simulate environments when solving the RL tasks. The idea to transfer status and policy from status domain to visual domain with domain adaptation is interesting.


This paper shows some interesting ideas and novelty. But it still contains some uncleared descriptions and concerns. There are a few questions or problems as listed below:

	1. L135 mentioned that related works learn the mapping function by minimizing the divergence between distributions. What related works exactly here are talking about? Is there any comparison of these works in your experiments?
	2. L146 What is the Gaussian distribution here descript? The structure of q() and the output distribution should be clarified.
	3. L158 The difference of ELBO loss with VAEGAN should be more specific list here.
	4. The definition of D in Eq4. is unclear.
	5. L178 What are the structure of the DM and embedded DM model? This is not clear.

Why name your method CODAS instead of CMDAS ?


**Time Spent Reviewing:**

4

---

> ### Author Response · Authors · 2021-08-09
> **Official Response to Reviewer cBxr**
>
> Thanks for your constructive comments on our paper. We will amend our paper for better clarity. Our answers to your questions are as follows:
>
>
> **1. This paper has limited impact on the related community since similar works have been explored.**
>
> We do not agree on this point.  We would like to argue about the novelty of CODAS.
>
> In this paper, we propose a new setting of cross-modal domain adaptation in RL. The motivation of cross-modal domain adaptation is to reduce the costs in current visual sim2real problems that require an image-based simulator. First, a rendering engine needs costly human engineering. Second, using RL methods to train a policy with an image-based simulator is usually harder and slower. As far as we know, this is the first paper that points out such a problem and proposes a new problem setting and its corresponding solution. We believe that efficient solutions built on a state-based simulator are valuable to improve the usability of UDA techniques in sim2real to more applications in the future.
>
> Besides, we point out that the distribution minimization objective is ill-posed for UDA in RL. Instead of adopting the distribution minimization objective and GAN directly as previous studies do, our algorithm is focused on rigorous derivations on unsupervised domain adaptation problems based on the framework of variational inference. We believe the new formulation on UDA of RL is valuable to inspire more future work to rethink their ultimate objective before building their UDA RL system.
>
> The differences and connections to existing works have been thoroughly discussed in the related work section of the main body and appendix (also see the answer to Q2 and Q5 below). We sincerely hope that the reviewer could re-evaluate the value of this paper.
>
>
>
> **2. What are the related works about distribution divergence minimization in L135?**
>
> The training objective of GAN is equivalent to the minimization of a certain distance measure between two distributions [1]. The mentioned image-to-image UDA algorithms in related work (L70 - 75) are based on this. In the experiments, we compare their modified image-to-state variants (see GAN and Cycle-GAN) with CODAS. We will update the description here for better clarity.
>
> **3. What does the Gaussian distribution in L146 describe?**
>
> The output of the model is the mean and standard error of a Gaussian distribution. We use a sampled value of this distribution (with reparameterization) as the input to the subsequent networks. We will add the description to the paper.
>
> **4. What are the structure of the DM and embedded DM model?**
>
>  The detailed model structure is illustrated by Fig. 2 in Appendix. We have also provided the model structure and hyperparameters in Sec. D.3.
>
> **5. What is the difference between CODAS and VAEGAN?**
>
> VAE-GAN uses an encoder to embed images into a latent space. It regards the decoder in the reconstruction loss as a generator and uses a discriminator to distinguish the reconstructed images from the real images in order to train a better encoder and decoder. It does not model sequential information. The motivation and the application scenarios are also different.
>
> **6. Why name your method CODAS instead of CMDAS?**
>
> We name it CODAS for easier pronunciation.
>
> [1]  Sebastian Nowozin, Botond Cseke, and Ryota Tomioka. "f-GAN: Training generative neural samplers using variational divergence minimization." NIPS 2016.

---

> > ### Comment · Reviewer_cBxr · 2021-09-01
> > **Reply to the response**
> >
> > Thanks for the comments and response. After carefully revisit the comments and other reviewers, I think the authors partially solved my concerns. However, I am still not convinced about the value of this exploration and its novelty comparing with other existing works. I think a specific section should be added to discuss the contribution of this work and compare it with related works. Also, figure 3 in the paper and figure 2. in the Appendix did not clearly show the structure of each network. Readers have to guess their structure via parameters in table 1. This cause the experiments to be not easy to follow. It seems this work contains some novelty that practically optimize the visual simulation problems but the theoretical contributions are not clearly introduced.
> > I believe this paper needs to be further polished to reveal its novelty in theory. I would like to keep my current score on this paper.

---

> > > ### Author Response · Authors · 2021-09-02
> > > **Response to Reviewer cBxr**
> > >
> > > Thanks for the in-time response.  However, we cannot agree with your comments on the following aspects. We would appreciate it if you could give us a more detailed comment because we are somehow confused by what you have mentioned.
> > >
> > > 1. The main contributions of our work are explicitly stated in the abstract, introduction, and discussion sections which include proposing a state-to-image UDA setting, pointing out the ill-posedness of previous UDA problems, and providing a practical solution to solve the problem. They are consistent throughout the paper. The comparison is also explicit in the related work section of the main body and appendix. In particular, as far as we know, previous methods all need renderable simulators and do not clearly discuss the ill-posedness problem in the UDA of RL.
> > >
> > > 2. The illustration in Fig. 2 of the Appendix is designed to emphasize the module-level structure of CODAS and its information flow so it does not contain a detailed layer-level structure. Since all the layers used in CODAS are standard CNN and MLP and our design is focused on the information flow, we choose not to include them in the illustration for better brevity and instead list their sizes and activation functions in Table 2. We will open-source the code to demonstrate the network design after the final decision.

---

### Official Review · Reviewer_NDFM · 2021-07-20

**Rating:** 8
**Confidence:** 3

**Summary:**

This paper studies the problem setting in which one has a "source" simulator with low-level state information in which one can efficiently train an agent to perform a task, and a "target" simulator that shares the initial state distribution and transition dynamics as the source simulator but only offers high-dimensional observations. The goal is to leverage trajectories sampled from a policy in the target domain to learn a mapping $q_\phi$ from the stream of high-dimensional observations in the target domain to states in the source domain. One can then perform the task in the target domain by feeding observations through $q_\phi$ and then through the source policy. The authors propose to use variational inference to train $q_\phi$ and thoroughly demonstrate what adaptation tasks their framework (CODAS) can and cannot perform on MuJoCo and Hand Manipulation. Impressively, using CODAS, the model in the target domain generally achieves 80-90% the performance of the model in the source domain.

**Limitations And Societal Impact:**

The authors do a good job of discussing the importance of the assumption in the derivation of CODAS that the dynamics and data collection policies match between source and target. They also offer an example of a task for which CODAS performs poorly (relocate in Table 1) and claim the reason is the "goal-conditional nature" of the task.

**Main Review:**

I enjoyed reading this paper. The problem domain is largely very practical; the assumption in 125-127 that the source and target policies are identical is somewhat limiting in that in the practical setting, we have an optimal policy in simulation but no optimal policy in the real world target domain. However, the authors do a good job of investigating this limitation and also providing intuition on what makes the alignment problem difficult even when it is assumed that the source policy matches the target policy. The experimental results are impressive and the writing is very clear. The algorithm does seem to be somewhat unstable and possibly difficult to train (based on section 3.3) but the authors provide a comprehensive description of their implementation details in the paper and supplementary.

Some minor comments below:

- Figure 1 - this figure seems to suggest that the "shape" of the 3-point trajectory is preserved from one domain to another, and that additional structure is what makes the problem easier. However, trajectory shapes in one domain do not necessarily correspond to the same shape in another domain. Is this interpretation and criticism of the figure correct?
- Figure 7 - is it also worth considering ablating the importance of the initial state distributions matching?
- 98 - how well is $q_\phi$ able to extract the correct state from an observation quantitatively? A training curve would be interesting to see. Figure 5 is just qualitative.
- 230-231 - what is the intuition for image->action requiring a higher sample complexity to learn than image->state?
- 245-247 - is there some suggestion for how one could fine-tune either the source policy or $q_\phi$ to bump the performance from 85% to 100%?
- 306-307 - is it not easy to collect a larger dataset and test this reasoning?

**Time Spent Reviewing:**

5

---

> ### Author Response · Authors · 2021-08-09
> **Official Response to Reviewer NDFM**
>
> Thanks for your appreciation of our paper and your constructive comments to it. Our answers to your questions are as follows:
>
> **1. What is the correct interpretation of Fig. 1?**
>
> The illustration of Fig. 1 does not suggest that the "shape" of trajectories in two domains should match. It is designed to demonstrate that with the context of "previous mapped states", we can rule out some wrong mappings. For example, $o_1$ in the left domain can be mapped to $s_1$ and $s_1'$ in the right domain using distribution matching. But if we know that $o_0$ has been mapped to $s_0$ instead of $s_0'$, we can then deduce that $o_1$ should be mapped to $s_1$ as well.  We will update the figure to disambiguate it.
>
> **2. Is it also worth considering ablating the importance of the initial state distributions matching?**
>
> Thanks for your suggestion on investigating the influence of the initial mapping. In MuJoCo environments, the initial states are initialized with perturbations, and the initial inferred state of CODAS is set to zero. In this implementation, CODAS itself seems to be robust to the randomness in the initial state. We have conducted experiments that assign the initial inferred states to the oracle initial state in the environment and cannot find significant improvement in performance.
>
> **3. How well is $q_\phi$ able to extract the correct state from an observation quantitatively?**
>
> The quantitative results are provided in Fig. 6 in Sec. F.2 of Appendix. CODAS consistently enjoys a lower mapping error compared with other methods.
>
> **4. What is the intuition for image$\to$action requiring a higher sample complexity to learn than image$\to$state?**
>
> Empirically, in Mujoco tasks, with compounding error in MDP, behavioral cloning needs a large dataset to recover a good-performance policy [1].  However, we do not think that learning an image$\to$action mapping always bears a higher sample complexity than learning an image$\to$state mapping does. The complexity is task-dependent.  For those tasks whose policy behavior is more complex than the relation between images and states, the technique of Unsupervised visual Domain Adaptation (UDA) is more sample efficient than behavioral cloning.
>
> **5. Are there some suggestions for how one could fine-tune either the source policy or $q_\phi$ to bump the performance from 85\% to 100\%?**
>
> Based on our experience, the potential solutions to further boost the performance may be:
>
> - incorporating more human knowledge into the structure mapping function, like how we designed the embedded-DM;
> - using a larger dataset of the target domain;
>
> **6. Is it not easy to collect a larger dataset and test this reasoning?**
>
> Loading a large dataset of states and images consumes massive memory. We are trying to optimize the memory usage to conduct the experiment.
>
> [1] Jonathan Ho and Stefanom Ermon. "Generative adversarial imitation learning." NIPS 2016.

---

> > ### Comment · Reviewer_NDFM · 2021-08-22
> > **Thank you for the detailed rebuttal**
> >
> > I appreciate the detailed rebuttal. There is great value in studying techniques for training agents in simulation and transferring them to the real world, and the framework for solving this problem that is introduced in this paper (e.g. train an agent to solve a task given state information, then learn to transform observations to states without access to (state, observation) pairs) is principled and practical. As pointed out by other reviewers, there are points of departure from the practical setting and the setting ultimately used for experiments in this paper (such as the use of identical source and target data collection policies and identical initial state distributions). However, I feel these restrictions are analyzed with clarity and honesty in the paper. I don't believe that it is necessary to demonstrate successful application of CODAS on a real-world task as requested by reviewer 6QJy - those experiments should be done eventually but I think it's fully justified to leave these for later work. I think my score is appropriate and I encourage the other reviewers to increase their scores.

---

> > > ### Author Response · Authors · 2021-08-26
> > > **Thanks for the response**
> > >
> > > We are glad that our response has solved your concerns. Many thanks for your appreciation to our work.

---

> ### Author Response · Authors · 2021-08-15
> **Experimental Results of CODAS Performance Under Initial-State Distribution Mismatch**
>
> In Hopper tasks, the initial state is generated by a constant state with additional perturbations from a uniform distribution $U(-\alpha, \alpha)$, where $\alpha=0.005$ in the original tasks.
>
> We further test the robustness (without oracle) to the initial state distribution mismatching by setting the initial inferred state to zero, and increasing $\alpha$ in the source domain.
>
> The performance of CODAS remains unchanged when $\alpha$ is doubled to 0.01. When $\alpha$ is increased to 0.1, CODAS can still reach a reward ratio of around 0.6, which is 82% of the original performance. The results prove that, without an oracle initial state mapping, CODAS is still robust to a stochastic initial state distribution under the current model design. However, if the initial state is indeed highly stochastic (e.g., $\alpha=0.5$), an oracle initial state mapping may help the performance.
>
> We will add the results to the revised paper. The original figure can be found in https://postimg.cc/qN05pDbR

---

### Author Response · Authors · 2021-08-30
**General Response**

Thanks for the appreciation and constructive comments of the reviewers on our work.

In the discussion period, we added results on initial-state distribution mismatching and data-size tolerability. We updated the results in the following anonymous links: 1. tolerance to initial-state distribution mismatching: https://postimg.cc/14QQY684; tolerance to smaller datasets: https://postimg.cc/p5Zks3pQ. We also responded to the reviewers’ other concerns about this work.

Based on the discussion, we believe that the proposed cross-modal UDA setting and the CODAS method which reformulate the UDA problem in RL and handle the ill-posed problem are valuable to inspire more powerful methods and applications to the community.

We think the previous responses have answered the reviewers' concerns about the work. Since there are no additional questions about the work, we sincerely hope the reviewers can consider increasing the score of this work. We also look forward to reviewers’ responses if you have any further concerns about this work.

---

### Decision · Program_Chairs · 2021-09-27

**Decision:**

Accept (Poster)

**Comment:**

This paper tackles the problem of visual-input sim-to-real and tries to overcome the reality gap between images rendered in simulators and those from the real world. It presents an approach that, instead of building and running simulators that render costly high-quality images, generates only low-dimensional states in simulation to learn the policy, and then learns encoders from and decoders to observations that handle sequential observations. This enables to train and transfer state dynamics from a source environment to a target environment with only high-dimensional observations. It then proposes a mathematical derivation for the cross-modal unsupervised domain adaptation problem. The method is then evaluated on MuJoCo tasks (OpenAI Gym and robot hand manipulation) and compared to several GAN, CycleGAN and temporally stacked GAN baselines, where it shows better domain adaptation and transfer performance.

Reviewer NDFM praised the paper and their minor comments were addressed. Reviewer cBxr’s most negative comment was about the limited impact of the work due to similarity with existing work, but the authors contest this review and justify their contribution as a combination of mathematical derivation of the problem as variational inference and the addition of sequential modeling of the dynamics (as opposed to VAE-GAN). Reviewer 6QJy complained about missing real-world data evaluations to make this work relevant (a claim disputed by the authors and reviewer NDFM), as well as missing comparisons to two recent sim2real techniques that “adapt image and state to hidden state and use that to train the policy'' and that modeled sequences of states from observations (the authors provide a clarification).

Review scores are (5, 5, 6, 8, average 6). Reviewers cBxr (score 5) did not respond to the rebuttal and did not update their scores but I believe they could promote their score. While waiting on this reviewer, I am therefore willing to challenge them and promote this paper to an acceptance.